# Upstream flow effects revealed in the EastGRIP ice core using Monte-Carlo inversion of a two-dimensional ice-flow model

Tamara Annina Gerber[1], Christine Schøtt Hvidberg[1], Sune Olander Rasmussen[1], Steven Franke[2], Giulia Sinnl[1], Aslak Grinsted[1], Daniela Jansen[2], and Dorthe Dahl-Jensen[1, 3]

[1]Section for the Physics of Ice, Climate and Earth, The Niels Bohr Institute, University of Copenhagen, Copenhagen, Denmark
[2]Alfred Wegener Institute, Helmholtz Centre for Polar and Marine Research, Bremerhaven, Germany
[3]Centre for Earth Observation Science, University of Manitoba, Winnipeg, Canada

**Correspondence:** Tamara Annina Gerber (tamara.gerber@nbi.ku.dk)

**Abstract.** The Northeast Greenland Ice Stream (NEGIS) is the largest active ice stream on the Greenland Ice Sheet (GrIS) and a crucial contributor to the ice-sheet mass balance. To investigate the ice-stream dynamics and to gain information about the past climate, a deep ice core is drilled in the upstream part of the NEGIS, termed the East Greenland Ice-Core Project (EastGRIP). Upstream flow can introduce climatic bias in ice cores through the advection of ice deposited under different conditions further upstream. This is particularly true for EastGRIP due to its location inside an ice stream on the eastern flank of the GrIS. Understanding and ultimately correcting for such effects requires information on the atmospheric conditions at the time and location of snow deposition. We use a two-dimensional Dansgaard–Johnsen model to simulate ice flow along three approximated flow lines between the summit of the ice sheet (GRIP) and EastGRIP. Isochrones are traced in radio-echo-sounding images along these flow lines and dated with the GRIP and EastGRIP ice-core chronologies. The observed depth-age relationship constrains the Monte-Carlo method which is used to determine unknown model parameters. We calculate backward-in-time particle trajectories to determine the source location of ice found in the EastGRIP ice core and present estimates of surface elevation and past accumulation rates at the deposition site. Our results indicate that increased snow accumulation with increasing upstream distance is predominantly responsible for the constant annual layer thicknesses observed in the upper part of the ice column at EastGRIP and the inverted model parameters suggest that basal melting and sliding are important factors determining ice flow in the NEGIS. The results of this study form a basis for applying upstream corrections to a variety of ice-core measurements, and the inverted model parameters are useful constraints for more sophisticated modelling approaches in the future.

## 1 Introduction

The East Greenland Ice-Core Project (EastGRIP) is the first attempt to retrieve a deep ice core inside an active ice stream. The drill site is located in the upstream part of the Northeast Greenland Ice Stream (NEGIS, Fahnestock et al., 1993), which is a

substantial contributor to the Greenland Ice Sheet (GrIS) mass balance (Khan et al., 2014) and accounts for around 12 % of its total ice discharge (Rignot and Mouginot, 2012). Large-scale ice-sheet models are essential tools to anticipate the future development of the NEGIS and its potential impact on the stability of the GrIS (Joughin et al., 2001; Khan et al., 2014; Vallelonga et al., 2014). However, results obtained from such models often show a significant deviation from observed surface velocitites in the NEGIS and its catchment area (Aschwanden et al., 2016; Mottram et al., 2019). In particular, the high ice-flow velocities in the upstream area of the NEGIS and the clearly defined shear margins are difficult to reproduce with ice-flow models (Beyer et al., 2018). A recent study by Smith-Johnsen et al. (2020a) showed, that the high surface velocities in the onset region of the ice stream could be reproduced with their model, using an exceptionally high and geologically unfeasible geothermal heat flux (Bons et al., 2021). This indicates that additional, yet unknown processes must facilitate ice flow in the NEGIS and that the driving mechanisms governing ice flow are still not understood well enough. The EastGRIP ice core sheds some light on the key processes by revealing unique information about ice dynamics, stress regimes, temperatures and basal properties, all of which are crucial components in ice-flow models.

Chemical and physical properties measured in ice cores reflect the atmospheric conditions at the time and location of snow deposition (e.g. Alley et al., 1993; Petit et al., 1999; Andersen et al., 2004; Marcott et al., 2014). Most of the deep drilling projects in Greenland and Antarctica are located in slow-moving areas at ice domes or near ice divides (e.g. GRIP (Dansgaard et al., 1982), Dome Fuji (Ageta et al., 1998), Dome C (Parrenin et al., 2007)), so the ice core can be expected to represent climate records from this fixed location. For ice cores drilled on the flank of an ice sheet (e.g. GISP2 (Meese et al., 1997), Vostok (Lorius et al., 1985; Petit et al., 1999)) or in areas with higher flow velocities (e.g. Camp Century (Dansgaard and Johnsen, 1969), Byrd (Gow et al., 1968), NorthGRIP (Andersen et al., 2004), EDML (Barbante et al., 2006), WAIS Divide (Fudge et al., 2013), NEEM (NEEM Community members et al., 2013)), the ice found at depth was originally deposited further upstream and advected with the horizontal flow.

The spatial variation in accumulation rate, surface temperature and atmospheric pressure can introduce climatic imprints in the ice-core record which stem from the advection of ice deposited under different conditions further upstream. The ice core signal is thus a combination of temporal and spatial variations in climatic components (Fudge et al., 2020). The magnitude of these so-called upstream effects depends on the ice-flow velocity, spatial variability of the precipitation and the sensitivity to atmospheric variations of the quantity under consideration. While well-mixed atmospheric gases, such as carbon dioxide or methane, and dry-deposited impurities are barely affected (Fudge et al., 2020), properties extracted from the ice phase can show significant bias. Affected measurements include aerosols and cosmogenic isotopes, such as $^{10}$Be (Yiou et al., 1997; Finkel and Nishiizumi, 1997; Raisbeck et al., 2007; Delaygue and Bard, 2011), the isotopic composition of water (Dansgaard, 1964; Jouzel et al., 1997; Aizen et al., 2006), the total air content (Raynaud et al., 1997; Eicher et al., 2016) and ice temperatures (Salamatin et al., 1998). Processes such as vertical thinning of the ice column and firn densification are also influenced by upstream effects and have consequences on the annual layer thicknesses (Dahl-Jensen et al., 1993; Rasmussen et al., 2006; Svensson et al., 2008) and the age difference between ice and the enclosed air (Herron and Langway, 1980; Alley et al., 1982). Upstream effects in the EastGRIP ice core are expected to be particularly strong due to the fast ice flow in the upstream area (57 ma$^{-1}$ at EastGRIP, Hvidberg et al., 2020), the strong gradient in the accumulation rate across Greenland's main ice ridge

(Burgess et al., 2010), and the increasing elevation towards the central ice divide (Simonsen and Sørensen, 2017). The correction of these effects in the EastGRIP ice core is necessary to interpret the ice-core measurements within the climatic context and requires information on the conditions at the time and location of snow deposition.

Post-depositional deformation of isochrones observed in radio-echo-sounding (RES) images along flow lines provides information on ice-flow dynamics and can be used to reconstruct past and present flow characteristics. In this study, we use a vertically two-dimensional Dansgaard–Johnsen model to simulate the propagation and deformation of isochrones along three approximated flow lines between the ice-sheet summit (GRIP) and EastGRIP. A Monte-Carlo method is used to determine the unknown model parameters by minimizing the misfit between modelled and observed data. The latter includes the depth of isochrones observed in RES images along the flow lines and a parameter $\alpha_{sur}$ representing the sum of the horizontal strain rates deduced from satellite based surface velocities. From the modelled velocity field, we calculate particle trajectories backwards in time to infer the source location of ice found in the EastGRIP ice core and estimate the accumulation rate at the time of snow deposition. The source characteristics presented here form a basis to correct for upstream effects in various chemical and physical quantities of the EastGRIP ice core. These corrections are important to remove any climatic bias in ice-core measurements which are currently analyzed and will become available in the future. The inverted model parameters give insight into basal properties and ice-flow dynamics along the flow lines and can be used to constrain more sophisticated numerical models of the NEGIS.

## 2   Data and methods

Snow layers deposited at the surface of ice sheets are buried with time and are deformed as a consequence of ice flow. While these isochrones can be observed in RES images today, the ice-flow characteristics which shaped them are generally unknown. This is a typical geophysical inverse problem and can be formulated as $\mathbf{d} = g(\mathbf{m})$, where the function $g(\mathbf{m})$ represents the ice-flow model linking the model parameters ($\mathbf{m}$) with the observed data ($\mathbf{d}$). A variety of inverse methods exist to find the model parameters which reproduce the observed data within their uncertainties. Here, we use a Markov Chain Monte-Carlo method to determine the unknown parameters of a two-dimensional ice-flow model by minimizing the misfit between modelled and observed isochrones and strain rates. This allows us to reconstruct the ice-flow characteristics in the past and to determine the flow trajectories of the EastGRIP ice.

In the coming sections we describe the data and methods underlying our results according to the work flow illustrated in Fig. 2. In Sect. 2.1 to 2.3 we explain how the isochrone depth-age relationship constraining the Monte-Carlo method was obtained. This involves the selection of RES images approximating the EastGRIP flow line (Sect. 2.1), extending the existing chronology of the EastGRIP ice core to the current drill depth (Sect. 2.2), and the tracing and dating of isochrones in the RES data (Sect. 2.3). In Sect. 2.4 the ice-flow model is described in detail and in Sect. 2.5 we elaborate on the Monte-Carlo method used for parameter sampling. The section numbers are displayed in the corresponding steps in Fig. 2.

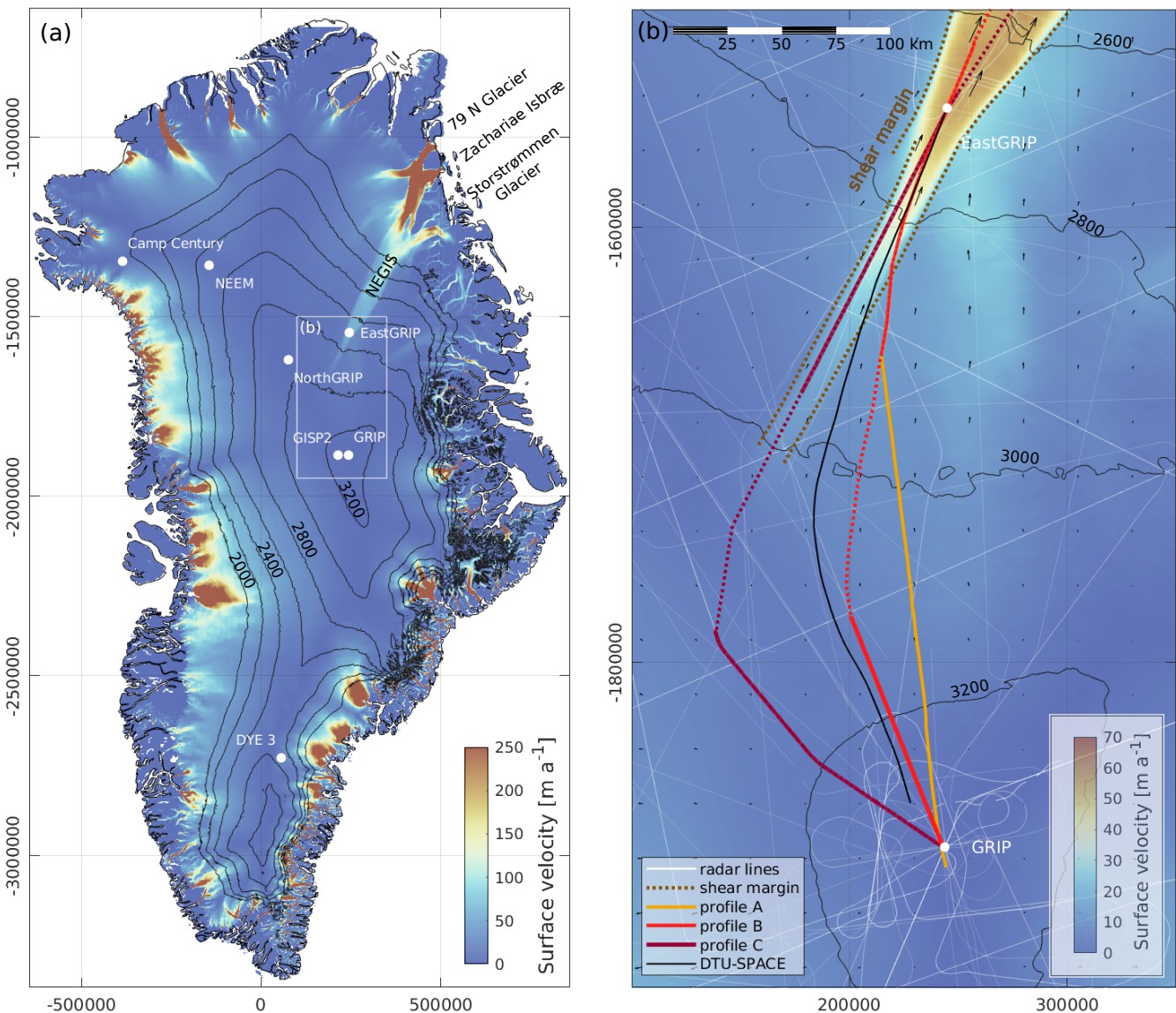

**Figure 1. (a)** Overview of past and ongoing deep ice-core drilling projects on the GrIS (surface elevation and Greenland contour lines by Simonsen and Sørensen, 2017; Greene et al., 2017) and the outline of the study area. The NEGIS appears as a distinct feature in the surface velocities (Joughin et al., 2018). It extends from the central ice divide to the northeastern coast, where it splits up into the three marine-terminating glaciers 79N Glacier, Zachariae Isbræ and Storstrømmen Glacier. **(b)** The present-day EastGRIP flow line is derived from the DTU_SPACE surface velocity product (Andersen et al., 2020). Due to the limited availability of radar data along the flow line, we construct three approximate flow lines through a combination of various radar products (profile A–C) between GRIP and EastGRIP. Flow line B and C lack data in the centre of the profiles, marked as a dashed line. The downstream parts of line A and B comprise the same radar profile, which crosses the southern shear margin around 82 km upstream of EastGRIP.

## 2.1 EastGRIP flow lines

Determining the exact flow line through the EastGRIP ice-core site is important to understand the flow history of the survey area and enables us to reconstruct the location where the ice from the ice core was deposited at the ice-sheet surface. For this, we use high-resolution satellite-based surface velocity products (e.g. Joughin et al., 2018; Gardner et al., 2020; Andersen et al., 2020, see supplementary material Fig. S1) to calculate the upstream flow path. Minor uncertainties and bias in these data products affect along-flow tracing and lead to deviations between flow lines derived from different velocity maps. These deviations become more pronounced with increasing distance from the starting point, as the uncertainties propagate along the line and in general become larger in slow-moving areas of the ice sheet (Hvidberg et al., 2020). Due to the small bias, we consider the DTU_SPACE (Andersen et al., 2020) line the most likely current flow line (Fig. 1b). Yet, there is no evidence that the present-day velocity field was the same in the past. A slight shift in the NEGIS shear margins or the central ice divide, for instance, would have a large effect on the velocity field and, hence, the determination of the flow line of the EastGRIP ice remains ambiguous.

RES data reveal the internal structure of glaciers and ice sheets and provide valuable information on the ice-flow characteristics, particularly when recorded parallel to the ice flow. The electromagnetic waves used in RES are sensitive to contrasts in dielectric properties of the medium they propagate. In ice sheets, these contrasts arise through density variations in the uppermost part of the ice column (Robin et al., 1969), changes in the crystal orientation fabric (Harrison, 1973), and impurity layers such as volcanic deposits (Paren and Robin, 1975). The latter is the most common reflector type below the firn (Millar, 1982; Eisen et al., 2006), and because it is related to layers deposited over a relatively short period of time, most internal reflection horizons (IRHs) detected by RES can be considered isochrones.

The availability of RES data in the study area is limited, and unfortunately, the flight lines generally do not follow the surface velocity field. We have thus composed three approximated flow lines connecting the EastGRIP (75.63° N, 35.99° W, 2720 m) and the GRIP (72.58° N, 37.63° W, 3230 m) drill sites from the available RES data sets (Fig. 1b). The radar data used in this study (Table 1) were measured by the Alfred Wegener Institute, Helmholtz Centre for Polar and Marine Research (AWI, Jansen et al., 2020; Franke et al., 2021b) and the Centre for Remote Sensing of Ice Sheets (CReSIS, 2020). The AWI data were recorded by an 8-antenna-element ultra-wideband radar system (MCoRDS5) mounted on the Polar 6 Basler BT-67 aircraft, operating at a frequency range of 180–210 MHz (Franke et al., 2020, 2021b). The CReSIS radar data were measured by a ICORDS 2 (1999) and a MCoRDS 2 (2012) radar system, mounted on a P3 aircraft, at a frequency range of 141.5–158.5 MHz and 180–210 MHz, respectively. Details of the three radar systems are provided in Table 2.

The downstream parts of profile A and B consist of the same flight line, which passes through the EastGRIP drill site and intersects the southern shear margin around 82 km upstream of EastGRIP. Outside the NEGIS, the two lines split up and connect to two different RES profiles. Line B remains relatively close to the flow direction of the DTU_SPACE line but has a wide data gap in the centre of the profile. In line A, this problem is circumvented by using a radar profile connecting directly to GRIP, which deviates from the observed surface flow field by more than 15 degrees at some locations. Profile C follows the NEGIS trunk all the way to the central ice divide and connects to GRIP over the ice ridge without crossing the shear margin. Similar

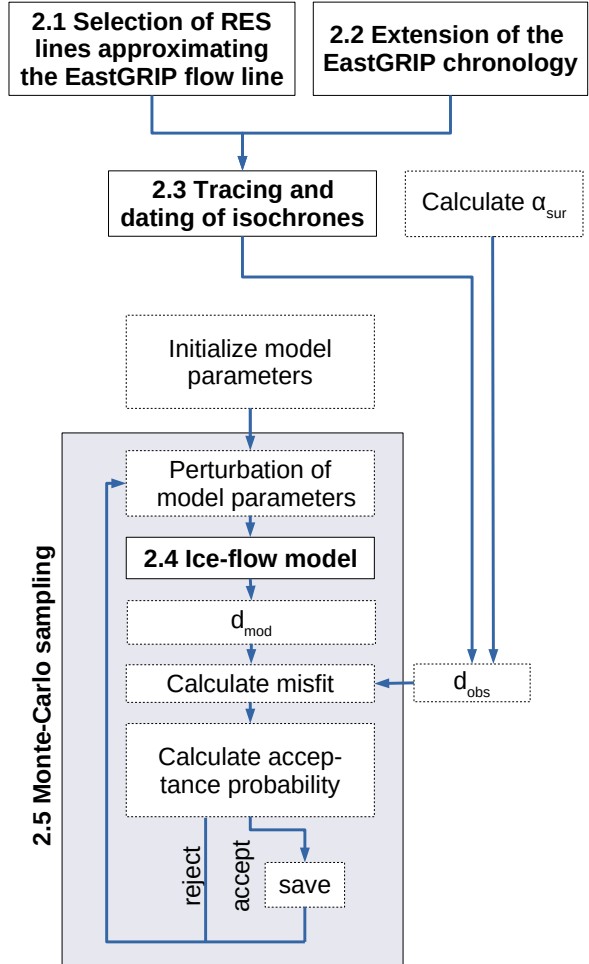

**Figure 2.** Workflow of the applied steps leading to the results described in Sect 3. The main steps are described in Sect. 2.1 to 2.5 and marked with the corresponding numbers in the figure: The observed data ($\mathbf{d}_{obs}$) constraining the Monte-Carlo method consists of the $\alpha_{sur}$ calculated from the ice surface velocities and the isochrone depths along the flow lines. The latter is obtained by approximating the EastGRIP flow line with selected RES images (Sect. 2.1), extending the EastGRIP chronology to the current drill depth (Sect. 2.2) and subsequent tracing and dating of isochrones (Sect. 2.3). The iterative Monte-Carlo sampling process is illustrated in the grey box (Sect. 2.5) and includes data simulation by a Dansgaard-Johnsen ice-flow model described in Sect. 2.4.

to flow line B, flow line C contains a substantial data gap between the onset region of the NEGIS and the central ice divide.

To avoid uncertainties related to the proximity of the model boundaries, the flow lines were extended more than 50 km beyond EastGRIP and have a total length of 422 (line A), 421 (line B) and 480 km (line C). To account for any differences in surface elevation or topography between RES data from different years, the ice surface reflections of the radar profiles were
125 aligned to the surface elevation from the Arctic DEM (digital elevation model, Porter et al., 2018). The bed topography in the data gaps of the profiles was derived from the BedMachine v3 data set (Morlighem et al., 2017).

**Table 1.** RES profiles used to approximate the EastGRIP flow lines A–C. The data sets were measured between 1999 and 2018 by CReSIS (CReSIS, 2020) and AWI (Jansen et al., 2020; Franke et al., 2021b).

| Flow line | Data files | Institution | Year | Radar system |
|---|---|---|---|---|
| A | Data_20180512_01_001 – 004 | AWI | 2018 | MCoRDS 5 |
| A | Data_19990512_01_009 – 010 | CReSIS | 1999 | ICoRDS 2 |
| B | Data_20180512_01_001 – 004 | AWI | 2018 | MCoRDS 5 |
| B | Data_19990523_01_016 – 017 | CReSIS | 1999 | ICoRDS 2 |
| C | Data_20180517_01_002 – 004 | AWI | 2018 | MCoRDS 5 |
| C | Data_20120330_03_008 – 011 | CReSIS | 2012 | MCoRDS 2 |

**Table 2.** Operating parameters of the radar systems used for data acquisition. Further details can be found in Gogineni et al. (2001), Byers et al. (2012) and Franke et al. (2021b).

| Parameter | ICORDS 2 | MCoRDS 2 | MCoRDS 5 |
|---|---|---|---|
| Bandwidth | 141.5–158.5 MHz | 180–210 MHz | 180–210 MHz |
| Tx power | 200 W | 1050 W | 6000 W |
| Waveform | Analogue chirp (SAW) | 8 channel chirp (2–3 waveforms) | 8 channel chirp (3 waveforms) |
| Sampling frequency | 18.75 MHz | 111 MHz | 1600 MHz |
| Transmit channels | 1 | 8 | 8 |
| Receiving channels | 1 | 15 | 8 |
| Range resolution | 7.6 m | 4.3 m | 4.3 m |

## 2.2 Extending the chronology of EastGRIP from GS-2 to GI-14

The validation of our modelling results and the correct dating of isochrones requires a reliable depth-age scale. The Greenland Ice Core Chronology 2005 (GICC05, Vinther et al., 2006; Rasmussen et al., 2006; Andersen et al., 2006; Svensson et al., 2006)
is based on annual layer counting in various Greenland ice cores. It has been transferred to GRIP and other deep drilling sites in Greenland by synchronizing the ice cores with each other using horizons of e.g. volcanic origin (Rasmussen et al., 2008;

Seierstad et al., 2014). The upper 1,383.84 m of the EastGRIP ice core were drilled between 2015 and 2018, and synchronized with the NorthGRIP ice core in previous work (Mojtabavi et al., 2020).

By 2019, the ice-core drilling progressed down to 2,122.45 m, allowing us to extend the existing time scale from 15 ka to 49.9 ka b2k (thousands of years before 2000 CE). As part of the present study, we identified common isochrones between EastGRIP, NorthGRIP and NEEM to transfer the GICC05 chronology to the part of the EastGRIP record which is not yet synchronized. This involved the same methods applied to NEEM by Rasmussen et al. (2013) and to the upper 1,383.84 m of EastGRIP by Mojtabavi et al. (2020). The isochrones chosen for synchronization purposes are mainly volcanic eruptions, which are registered as brief spikes in the electrical conductivity measurements (ECM, Hammer, 1980). The search of common ECM spikes was performed manually with a strong focus on finding patterns of similarly spaced eruptions rather than single and isolated events. The Matlab program 'Matchmaker' was used to visualize long data stretches and to evaluate the quality of the match (Rasmussen et al., 2008). An iterative multi-observer protocol was applied to reduce problems with confirmation bias and to ensure the reproducibility of the match.

A total of 138 match points were identified between 1,383.84 m and 2,117.77 m, adding to the previously known 381 match points. The match points between EastGRIP and the other two cores are shown in Fig. 3, representing all the volcanic tie points. The GICC05 chronology was transferred to EastGRIP by linear interpolation of depths between the match points. The age of the 1,383.84 m match point was already established to be 14,966 years b2k, which is near the termination of Greenland Stadial 2 (GS-2), with a reported maximum counting error (MCE) of 196 years (Mojtabavi et al., 2020). The age of the deepest match point was established to be 49,909 years b2k, just at the end of Greenland Interstadial 14 (GI-14), with an MCE of 2,066 years.

As in earlier similar work (e.g Rasmussen et al., 2013; Seierstad et al., 2014), very few match points were observed in the stadials, most clearly seen in Fig. 3 in the long stadial stages of GS-2 and GS-3. The sparse volcanic signals within stadial periods should not be attributed to a diminished global volcanic activity but rather to increased deposition of alkaline dust that neutralizes volcanic acid, caused by the prevailing colder and drier climatic conditions (Rasmussen et al., 2013). The largest distance between match points was observed across GS-2 and GS-3 and spans about 162 m of EastGRIP ice.

## 2.3 Tracing and dating of isochrones

The depth-age relationship from ice-core chronologies can be extended in the lateral plane by tracing and dating of isochrones in RES images. The depth of these isochrones along the EastGRIP flow lines is part of the observed data used to tune the ice-flow model parameters in the Monte-Carlo inversion. We traced 15 continuous IRHs and one non-continuous reflector along each of the three approximated flow lines with a semi-automatic Matlab program called 'picking tool'. The algorithm is based on calculating the local slope in each pixel of the RES image, and layers are traced automatically between two user-defined points. Starting from each of these points the algorithm walks along the steepest slope towards the other point. Subsequently, the two lines are weighted by distance to their starting point and combined to one layer. The number of picks required for thorough tracing depends on the data quality and reflector strength.

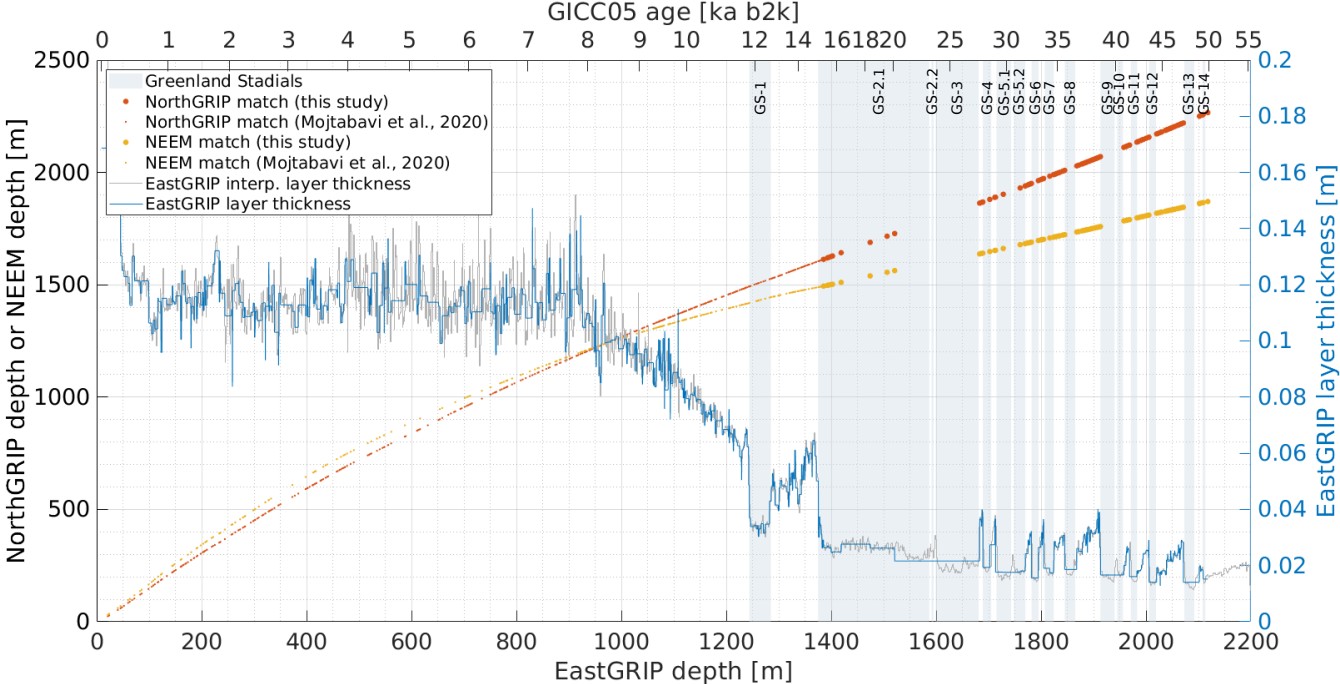

**Figure 3.** Synchronization between the EastGRIP, NorthGRIP and NEEM ice cores and comparison of match points obtained in this study with earlier results from Mojtabavi et al. (2020). The annual layer thickness of EastGRIP was computed after transferring GICC05 ages by linear interpolation to the EastGRIP ice core. The blue curve shows the annual layer thickness obtained by the match points only. The grey line indicates a high-resolution estimate of annual layer thicknesses at EastGRIP, obtained from the linear interpolation between the EastGRIP–NorthGRIP match points and assigning the interpolated EastGRIP depths to the NorthGRIP ages.

The total depth uncertainty ($\tilde{z}_t$) was calculated as

$$\tilde{z}_t = \sqrt{\tilde{z}_p^2 + \tilde{z}_{rr}^2}, \tag{1}$$

where the depth uncertainty introduced during the picking process ($\tilde{z}_p$) is estimated to be 10 m. The uncertainty related to the radar range resolution ($\tilde{z}_{rr}$) of the corresponding RES image is defined as

$$\tilde{z}_{rr} = \frac{kc}{2B\sqrt{3.15}}, \tag{2}$$

where $k$ is the window widening factor of 1.53, $c$ is the speed of light, $B$ is the radar bandwidth and 3.15 is the dielectric permittivity of ice.

The traced IRHs were dated at both drill sites by assigning the reflector depth at GRIP and EastGRIP to the corresponding time scale. In doing so, local irregularities were smoothed out by averaging the depth over $\pm 250$ m around the trace closest to the ice-core location. Because the EastGRIP ice core has not reached the bed yet, we extrapolated the time scale at EastGRIP with two IRHs observed below the current borehole depth to obtain a tentative depth–age relationship between 2,117.77 m and

175 the expected bed depth of 2,668 m.

The total age uncertainty ($\tilde{a}_t$) was estimated by following the approach described in MacGregor et al. (2015), where

$$\tilde{a}_t = \sqrt{\tilde{a}_c^2 + \tilde{a}_{rr}^2 + \tilde{a}_p^2} \tag{3}$$

takes into account the age uncertainties associated with the time scale ($\tilde{a}_c$, equivalent to 0.5 MCE), the radar range resolution ($\tilde{a}_{rr}$), and the layer picking process ($\tilde{a}_p$). The uncertainties related to the range resolution are estimated with

$$180 \quad \tilde{a}_{rr} = \frac{1}{2} \sum |a_c(z \pm \tilde{z}_{rr}) - a_c(z)|, \tag{4}$$

where $a_c$ is the ice-core age from the GICC05 time scale. Similar to Eq. (4), $\tilde{a}_p$ is estimated with

$$\tilde{a}_p = \frac{1}{2} \sum |a_c(z \pm \tilde{z}_p) - a_c(z)|. \tag{5}$$

The chosen isochrones show distinct patterns which could be identified in all RES images and allowed us to trace isochrones across disruptions and data gaps. Comparison of the isochrone depths at the ice-core locations obtained from different RES
images permits assessment of the quality of the tracing procedure. The high resolution of the radar images recorded in 2018 facilitates isochrone tracing, and the EastGRIP depths obtained from the two different AWI radar profiles agree to within 1.5 m. At GRIP, the discrepancy between isochrone depths obtained from three different radar profiles can be up to 30 m, which is slightly above the combined depth uncertainty related to the picking process and the resolution of the RES images. Lower range resolution and signal-to-noise ratio in older RES data introduce bias in isochrone identification, and although distinct
isochrones were chosen, a miscorrelation between IRHs recorded by different radar systems can not be entirely excluded. Moreover, the CReSIS profiles do not precisely intersect at GRIP and deviate from each other. The radar traces closest to GRIP are thus found at slightly different locations for the three RES images, which explains the higher discrepancy of radar layer depths.

The isochrone dating was conducted for each profile individually, and the obtained depths, ages and uncertainties were
195 averaged over the three lines (Table 3). The deepest non-continuous layer which could be identified at EastGRIP is found at a depth of $2,360 \pm 11$ m and is estimated to be $72,400 \pm 1,306$ years old. The layer depths of the continuously traced IRHs range from $421 \pm 11$ to $2,152 \pm 11$ m at the EastGRIP location, corresponding to ages of $3,498 \pm 94$ to $51,920 \pm 1,240$ years b2k. Reflectors 1–9 were deposited during the Holocene. The remaining reflectors are found in ice from the Last Glacial Period from which reflector 10 and 11 can be attributed to the onset of the Younger Dryas and the Bølling–Allerød. The relation between
200 the GRIP and EastGRIP depths of the traced IRHs fits well with the GICC05 time scale (Mojtabavi et al., 2020; Rasmussen et al., 2014), and the ages obtained from the two drill sites agree within the uncertainties. We note that the layer dating at EastGRIP consistently leads to younger ages than the dating at GRIP, which is a likely consequence of inaccuracies related to the transformation between ice-core and radar depths.

Due to computational reasons, we did not use all 16 layers for the Monte-Carlo inversion but picked ten isochrones with
205 approximately equal vertical spacing, and used the EastGRIP ages for our simulation of layer propagation. The layers used for the Monte-Carlo simulation are indicated in bold in Table 3 and plotted with a consistent color-code in Fig. 4, Fig. 5 and Fig. 7, representing the corresponding ages.

**Table 3.** Characteristics of the traced isochrones connecting the GRIP and EastGRIP ice-core sites. Displayed depths and ages are the average over the three flow lines. Depth uncertainties include the uncertainty related to the picking process and to the radar range resolution. Age uncertainties are related to the GICC05 time-scale uncertainties and isochrone depths. Figure 7d illustrates the depth and climatic context of these layers in the EastGRIP ice core, identified with the corresponding layer numbers. The bold layers and the EastGRIP ages were used for the Monte-Carlo inversion and are illustrated with a consistent color-code in Fig. 4, Fig. 5 and Fig. 7.

| Layer | GRIP depth [m] | EastGRIP depth [m] | GRIP age [yrs b2k] | EastGRIP age [yrs b2k] |
|---|---|---|---|---|
| 1 | $733 \pm 13$ | $421 \pm 11$ | $3,618 \pm 73$ | $3,498 \pm 94$ |
| **2** | **$795 \pm 13$** | **$471 \pm 11$** | **$4,004 \pm 74$** | **$3,945 \pm 95$** |
| 3 | $925 \pm 13$ | $573 \pm 11$ | $4,885 \pm 85$ | $4,805 \pm 93$ |
| **4** | **$1,217 \pm 13$** | **$838 \pm 11$** | **$7,178 \pm 106$** | **$7,139 \pm 95$** |
| 5 | $1,262 \pm 13$ | $882 \pm 11$ | $7,575 \pm 107$ | $7,531 \pm 95$ |
| **6** | **$1,347 \pm 13$** | **$968 \pm 11$** | **$8,364 \pm 122$** | **$8,321 \pm 110$** |
| 7 | $1,374 \pm 13$ | $996 \pm 11$ | $8,637 \pm 124$ | $8,600 \pm 113$ |
| 8 | $1,533 \pm 13$ | $1,153 \pm 11$ | $10,407 \pm 162$ | $10,365 \pm 149$ |
| **9** | **$1,592 \pm 13$** | **$1,208 \pm 11$** | **$11,209 \pm 181$** | **$11,140 \pm 168$** |
| 10 | $1,663 \pm 13$ | $1,282 \pm 11$ | $12,891 \pm 327$ | $12,822 \pm 290$ |
| **11** | **$1,749 \pm 13$** | **$1,355 \pm 11$** | **$14,612 \pm 281$** | **$14,350 \pm 206$** |
| **12** | **$2,039 \pm 13$** | **$1,704 \pm 11$** | **$28,633 \pm 840$** | **$28,522 \pm 647$** |
| **13** | **$2,193 \pm 13$** | **$1,903 \pm 11$** | **$38,015 \pm 994$** | **$37,914 \pm 793$** |
| **14** | **$2,298 \pm 13$** | **$2,035 \pm 11$** | **$45,463 \pm 1,189$** | **$45,174 \pm 1,086$** |
| **15** | **$2,395 \pm 13$** | **$2,152 \pm 11$** | **$52,602 \pm 1,360$** | **$51,920 \pm 1,240$** |
| 16 | - | $2,360 \pm 11$ | - | $72,400 \pm 1,306$ |

## 2.4 Ice-flow model

A full simulation of ice flow in the catchment area of the NEGIS is a highly under-determined problem (Keisling et al., 2014), lacking geophysical, climatic and ice-core data, some of which will become available in the future. Simpler models do not solve the problem in detail and are thus computationally much cheaper. Hence, limited but still useful information can be obtained from a simplified treatment of ice flow (e.g. Dansgaard and Johnsen, 1969; Dahl-Jensen et al., 2003; Waddington et al., 2007; Christianson et al., 2013; Keisling et al., 2014).

Here, we use a two-dimensional Dansgaard–Johnsen model (Dansgaard and Johnsen, 1969) to simulate the propagation and deformation of internal layers along approximated flow lines between the ice-sheet summit (GRIP) and EastGRIP. The simplicity of the model makes it well suited for the Monte-Carlo method due to its few model parameters, the allowance for large time steps, and because it has an analytical solution (Grinsted and Dahl-Jensen, 2002). The model assumes ice incompressibility and a constant vertical strain rate down to the so-called kink height ($h$) below which the strain rate decreases linearly. Basal sliding and melting are included in the model, and the ice-sheet thickness ($H$) is assumed to be constant in time.

We consider a coordinate system where the x-axis points along the approximated flow line, the y-axis is horizontal and perpendicular to the flow line, and the z-axis indicates the height above the bed. The horizontal velocities parallel ($u_{\parallel}$) and perpendicular ($u_{\perp}$) to the profiles are described by Grinsted and Dahl-Jensen (2002) as:

$$u_{\parallel}(z) = \begin{cases} u_{\parallel,sur}(x,y)\left[(1-f_{bed})\frac{z}{h} + f_{bed}\right], & z \in [0,h] \\ u_{\parallel,sur}(x,y), & z \in [h,H], \end{cases} \tag{6}$$

$$u_{\perp}(z) = \begin{cases} u_{\perp,sur}(x,y)\left[(1-f_{bed})\frac{z}{h} + f_{bed}\right], & z \in [0,h] \\ u_{\perp,sur}(x,y), & z \in [h,H] \end{cases} \tag{7}$$

where $u_{\parallel,sur}$ and $u_{\perp,sur}$ are the surface velocities parallel and perpendicular to the profile, and the basal sliding factor $f_{bed}$, is the ratio between the ice velocity at the bed and at the surface.

Ice flow in the vicinity of an ice stream is affected by lateral compression and longitudinal extension, in particular across the shear margins of the NEGIS. We thus introduce $\alpha = \frac{\partial u_{\parallel}}{\partial x} + \frac{\partial u_{\perp}}{\partial y}$ as the sum of the horizontal strain rates. Due to ice incompressibility we can write $\alpha + \frac{\partial \omega}{\partial z} = 0$, where $\omega$ symbolizes the vertical velocity. The x and y dependency in Eq. (6 - 7) only relates to the surface velocity, such that $\alpha_{sur}$ represents the horizontal dependency in the equations and can be calculated from the ice surface velocities.

The vertical velocities ($\omega$) are obtained through integration of the incompressibility relation $\omega(z) = -\int \alpha dz$ (Dansgaard and Johnsen, 1969):

$$\omega(z) = \begin{cases} \omega_{bed} - \alpha_{sur}(f_{bed}z + \frac{z^2}{2h}(1-f_{bed})) & z \in [0,h] \\ \omega_{sur} + \alpha_{sur}(H-z) & z \in [h,H]. \end{cases} \tag{8}$$

The boundary conditions for the vertical velocity at the bed ($\omega_{bed}$) and surface ($\omega_{sur}$) are

$$\omega_{bed} = -\dot{b} + f_{bed}u_{sur}\frac{\partial E_{bed}}{\partial x} \tag{9}$$

$$\omega_{sur} = -\dot{a} + u_{sur}\frac{\partial E_{sur}}{\partial x}, \tag{10}$$

where $\dot{b}$ is the positive basal melt rate, $\dot{a}$ is the positive accumulation rate, and $E_{bed}$ and $E_{sur}$ are the bed and surface elevations respectively. From Eq. (8) we derive the following expression for the modelled $\alpha_{sur}$:

$$\alpha_{sur} = \frac{\omega_{bed} - \omega_{sur}}{H - \frac{h}{2}(1-f_{bed})}. \tag{11}$$

Following Grinsted and Dahl-Jensen (2002) and Buchardt and Dahl-Jensen (2007), we adjust the accumulation rates and surface velocities to the climate conditions of the corresponding time with a scaling factor $\xi(t)$:

$$\xi(t) = e^{\kappa_2(\delta^{18}O - \delta^{18}O_w) - \frac{1}{2}\kappa_1(\delta^{18}O^2 - \delta^{18}O_w^2)}, \tag{12}$$

with $\quad \kappa_1 = \dfrac{c_w - c_c}{\delta^{18}O_w - \delta^{18}O_c}, \quad$ and $\quad \kappa_2 = c_w - \delta^{18}O_w\kappa_1.$

We use the oxygen isotope $\delta^{18}O$ record from NorthGRIP (Andersen et al., 2004) due to its high temporal resolution, and $\delta^{18}O_w = -35.2\,\permil$ and $\delta^{18}O_c = -42\,\permil$ are typical isotope values for warm interstadial and cold stadial periods, respectively. The parameters $c_w$ and $c_c$ determine the sensitivity of the accumulation rate with varying $\delta^{18}O$ in warm ($c_w$) and cold ($c_c$) periods and are defined as (Grinsted and Dahl-Jensen, 2002; Buchardt and Dahl-Jensen, 2007):

$$c_w = \frac{1}{\dot{a}}\frac{\partial \dot{a}}{\partial \delta^{18}O}\bigg|_{\delta^{18}O = \delta^{18}O_w}, \qquad c_c = \frac{1}{\dot{a}}\frac{\partial \dot{a}}{\partial \delta^{18}O}\bigg|_{\delta^{18}O = \delta^{18}O_c}. \tag{13}$$

To simulate the propagation of ice particles deposited at the surface of the GrIS, Eq. (6) and Eq. (8) are solved at a time interval of 10 years.

## 2.5 Monte-Carlo sampling

The ice-flow parameters $\dot{a}$, $h$, $f_{bed}$, and $\dot{b}$ are defined for intervals of $\sim$10 km along the flow lines, and form together with the two climate scaling factors, $c_c$ and $c_w$, the model vector $\mathbf{m}$. This results in a total of 170 (flow line A and B) and 194 (flow line C) model parameters. Each combination of them represents a possible solution to the inverse problem $\mathbf{d} = g(\mathbf{m})$, where $g(\mathbf{m})$ represents the ice-flow model described in the previous section. The data vector $\mathbf{d}$ contains the isochrone depths and $\alpha_{sur}$ determined from the MEaSUREs Multi-year v1 surface velocities (Joughin et al., 2018) at a resolution of one kilometer.

Like in most geophysical inverse problems, many different combinations of model parameters can explain the observed data equally well within the range of their uncertainties and therefore, a non-unique solution does not exist. Probabilistic inverse methods consider many different models and describe them in terms of their plausibility, rather than finding one possible solution. This makes these methods particularly well suited for nonlinear problems, where the probability density in the model space typically shows multiple maxima (Mosegaard and Tarantola, 1995).

Monte-Carlo methods are based on a random number generator which allows the sampling according to the target probability distribution in an efficient way. The grey box in Fig. 2 illustrates the iterative sampling process of the Metropolis algorithm (Metropolis et al., 1953) used here: Starting from an initial model ($\mathbf{m}_0$), a random walker explores the model space and proposes new models ($\mathbf{m}_{new}$) which are accepted with a certain probability ($P_{accept}$). This way of importance sampling avoids unnecessary evaluation of model parameters in low-probability areas (Mosegaard, 1998).

To estimate the initial accumulation rate $\dot{a}_0$, we integrate Eq. (8) (see Appendix A) and obtain the following depth–age relationship

$$(H - z) = \frac{\dot{a}}{\alpha_{sur}}(1 - e^{-\alpha_{sur}t}), \tag{14}$$

where $t$ and $z$ represent the isochrone age and height above the bed, respectively. The accumulation rate $\dot{a}$ is determined with a curve-fitting function, using at least 5 isochrones younger than 10 ka at each point along the flow line. The initial kink height

$(h_0)$, basal sliding $(f_{bed,0})$ and basal melt rate $(\dot{b}_0)$ are scaled with the normalized surface velocity $(\hat{\mathbf{u}}_{sur})$ as follows:

$$h_0 = H \left( \frac{1}{2} - e_1 \hat{\mathbf{u}}_{sur} \right) \tag{15}$$

$$f_{bed,0} = e_2 \hat{\mathbf{u}}_{sur} \tag{16}$$

$$\dot{b}_0 = e_3 \hat{\mathbf{u}}_{sur}, \tag{17}$$

where $e_1 = 0.4$, $e_2 = 0.8$ and $e_3 = 0.03$, and the initial value for $c_w$ and $c_c$ is assumed to be 0.15 and 0.10 respectively.

In each iteration a new model $\mathbf{m}_{new}$ is proposed as

$$\mathbf{m}_{new} = \mathbf{m}_0 + \mathbf{q}\mathbf{A}, \tag{18}$$

where $\mathbf{m}_0$ is the initial model and $\mathbf{A}$ contains the perturbation amplitude of the corresponding model parameter. The vector $\mathbf{q}$ defines the random walk in the multidimensional model space and solely depends on the preceding step. In each iteration, $i$, one model parameter, $j$, is randomly selected and perturbed as

$$\mathbf{q}_{i+1}(j) = \mathbf{q}_i(j) + \left( r - \frac{1}{2} \right) \mathbf{p}(j), \tag{19}$$

where $r$ indicates a random number between 0 and 1, and $\mathbf{p}$ regulates the maximum step length per iteration of the selected parameter type. To achieve a good performance of the Monte-Carlo algorithm, the values of $\mathbf{A}$ and $\mathbf{p}$ (shown in Table 4) are chosen such that the acceptance ratio for the individual model parameters lies between 25 % and 75 %.

The quality of the proposed model is evaluated by the function $S(\mathbf{m})$ which describes the misfit between the modelled and observed data (see Appendix B). The new model $(\mathbf{m}_{new})$ is accepted with the acceptance probability (Metropolis et al., 1953)

$$P_{accept} = \min \left( \frac{L(\mathbf{m}_{new})}{L(\mathbf{m}_{old})}, 1 \right), \tag{20}$$

where $\mathbf{m}_{old}$ is the last accepted model and the likelihood function is defined as $L(\mathbf{m}) = e^{-S(\mathbf{m})}$.

To ensure that parameter sampling is occurring in a physically reasonable range, the a priori probability distribution is assumed to be uniform within the following intervals:

$$\dot{a} \in \left[ \dot{a}_0 - 0.02 \mathrm{ma}^{-1}, \dot{a}_0 + 0.02 \mathrm{ma}^{-1} \right] \tag{21}$$

$$h \in [0, H] \tag{22}$$

$$f_{bed} \in \left[ \max(0, f_{bed,0} - 0.3), \min(1, f_{bed,0} + 0.3) \right] \tag{23}$$

$$\dot{b} \in \left[ 0 \mathrm{ma}^{-1}, 0.2 \mathrm{ma}^{-1} \right]. \tag{24}$$

The sampling intervals are based on expected values of the corresponding parameter: The initial accumulation rate obtained from the radar stratigraphy is considered to be quite trustworthy but because the local layer approximation is not justified in the survey area (Waddington et al., 2007) we allow the accumulation rate to deviate by $0.02~\mathrm{ma}^{-1}$. The kink height is limited to the ice-sheet thickness, the basal sliding fraction is allowed to deviate by 30 % from the initial model, and the upper limit of

the basal melt rate is based on values suggested at EastGRIP by a recent study (Zeising and Humbert, 2021).

In their initial phase, Markov Chain Monte-Carlo methods move from the starting model towards a high-probability area where the target distribution is sampled. To avoid sampling during this so-called burn-in phase, the first $1 \times 10^6$ accepted models are discarded. Since only one parameter is perturbed at a time, successive models are highly correlated. To obtain a distribution of independent models, only every 1000th accepted model is saved. The sampling is continued for $6 \times 10^6$ iterations in total.

**Table 4.** Perturbation amplitude (**A**) and step length (**p**) of the individual model parameters used for the Monte-Carlo sampling. The sampling parameters were chosen such that the acceptance ratio of the individual model parameters lies between 25 % and 75 %.

| Model parameter | Amplitude (A) | Step length (p) |
|:---:|:---:|:---:|
| $\dot{a}$ | 0.01 ma$^{-1}$ | 0.5 |
| $\dot{b}$ | 0.01 ma$^{-1}$ | 1 |
| $h$ | 100 m | 3 |
| $f_{bed}$ | 0.05 | 2 |
| $c_c$ | 0.05 | 0.5 |
| $c_w$ | 0.05 | 0.5 |

## 3 Results

### 3.1 Model parameters

Due to the mixed-determined nature of the inverse problem addressed in this study, a unique solution of model parameters does not exist. The Monte-Carlo sampling results in a number of possible models distributed according to the posterior probability. Here, we present the mean model parameters with the standard deviations of the posterior probability distribution and emphasize that the corresponding histograms (Fig. 6) are essential to understand the uncertainties of the parameter considered.

The flow-line characteristics and model parameters for each flow line are summarized in Fig. 4. The radar profiles with the observed and modelled isochrones are displayed as a function of the distance from the EastGRIP ice-core location. Particle trajectories were calculated from the simulated velocity field with the mean model parameters and indicate the source location of ice found at the modelled isochrone depth in the EastGRIP ice core. The isochrones and particle trajectories are illustrated with the same color-code as in Fig. 5 and Fig. 7f, indicating the corresponding age. The horizontal strain rates ($\dot{\varepsilon}_{xx}$, $\dot{\varepsilon}_{yy}$ and $\dot{\varepsilon}_{xy}$) were obtained from the MEaSUREs Multi-year v1 surface velocity components (Joughin et al., 2018) parallel ($u_{\parallel}$) and perpendicular ($u_{\perp}$) to the approximated flow line. The strain rates show mostly low, positive values along the flow lines with the exception of the shear-margin crossing in profile A and B, which is characterized by longitudinal extension and lateral compression.

The central observed features are the following:

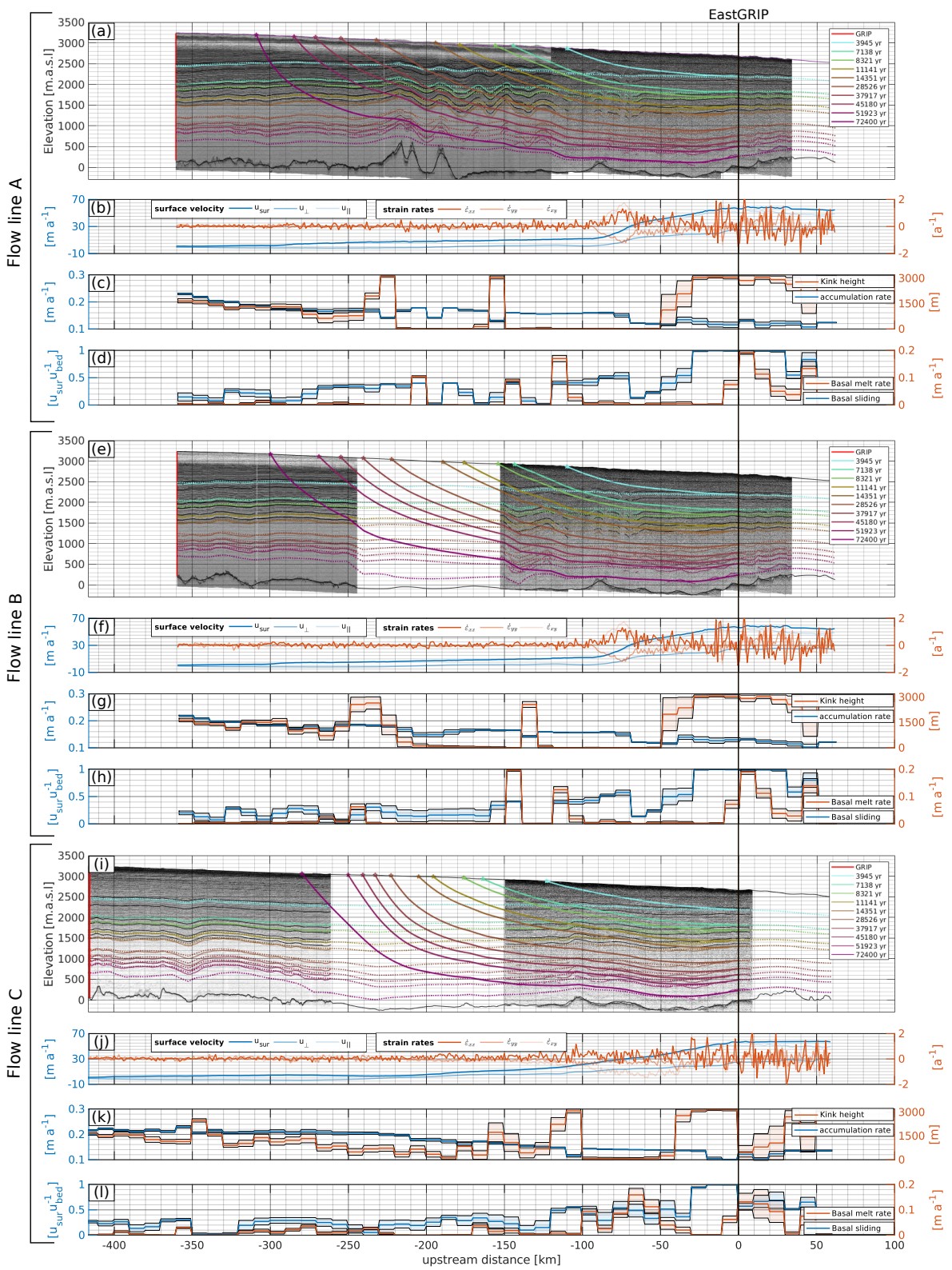

**Figure 4. (Previous page.)** Flow-line characteristics and model parameters for the approximated flow lines A **(a–d)**, B **(e–h)**, and C **(i–l)**. IRHs were traced (thin solid lines) in RES images and simulated (dashed lines) with a two-dimensional Dansgaard–Johnsen model **(a, e, i)**. From the modelled velocity field, we calculated particle trajectories (thick solid lines) backwards in time to obtain estimates of the source location for specific depths in the EastGRIP ice core. The colors of the lines indicate the age of the isochrones and the respective time of snow deposition and are identical to the color code in Fig. 5 and Fig. 7. The horizontal strain rates at the surface were calculated from the MEaSUREs Multi-year v1 (Joughin et al., 2018) surface velocities **(b, f, j)**. The mean and standard deviations of the sampled model parameters accumulation rate, kink height, basal melt rate and basal sliding **(c, d, g, h, k, l)** were obtained from a Monte-Carlo inversion by reducing the misfit between observed and simulated data. All panels are aligned at EastGRIP and the x-axis indicates the distance from the borehole location.

1. The accumulation rate decreases with increasing distance from the central ice divide. In flow line A and B, it remains almost constant between -220 and -80 km, followed by a drop of about 20 % across the shear margins. In the first ∼150 km of flow line C, which corresponds to the ice divide, the accumulation rate remains nearly constant, followed by a gradual decrease with increasing distance along the profile.

2. The kink height fluctuates around the middle of the ice column in the vicinity of the ice ridge and is drawn closer to the bed in the centre of the profiles. Locally very high kink heights are observed in flow line A around -230 km and -150 km, in flow line B at -240 km and -140 km, and at -100 km in flow line C. In all profiles, $h$ increases substantially at about -60 km.

3. The basal velocity ranges between 0 and 50 % of the surface velocity outside the NEGIS and increases to 60–100 % in the vicinity of EastGRIP.

4. The basal melt rate in the beginning of the profiles varies between 0 and 0.03 $\mathrm{ma}^{-1}$. As for the kink height, flow line A shows strong melt rate fluctuations in the centre of the profile, some of which are also observed in flow line B. At EastGRIP, basal melt rates between 0.05 and 0.1 $\mathrm{ma}^{-1}$ are obtained but higher values of up to 0.2 $\mathrm{ma}^{-1}$ are reached further downstream.

### 3.2 Monte-Carlo performance

The comparison of modelled and observed isochrones (Fig. 5a,c,e) and $\alpha_{sur}$ (supplementary material, Fig. S2) shows a good fit in most parts of the flow lines. However, our model is not able to accurately reproduce strong internal layer undulations which are not related to the bed topography or the surface conditions, resulting in a larger misfit where such undulations are present (Fig. 5b,d,f). In general, the isochrone misfit tends to be larger for deeper layers. Particularly distinct is the positive misfit at EastGRIP for the deepest layer in all profiles, indicating that the depths of old layers are overestimated. The average isochrone misfit for flow line A, B and C is 2.94 %, 2.34 % and 1.49 % of the respective layer depth.

Histograms in Fig. 6 show the sampled probability distribution of model parameters at GRIP and EastGRIP with the corresponding mean and standard deviation displayed on top. Distributions with distinctive single peaks and low standard deviation

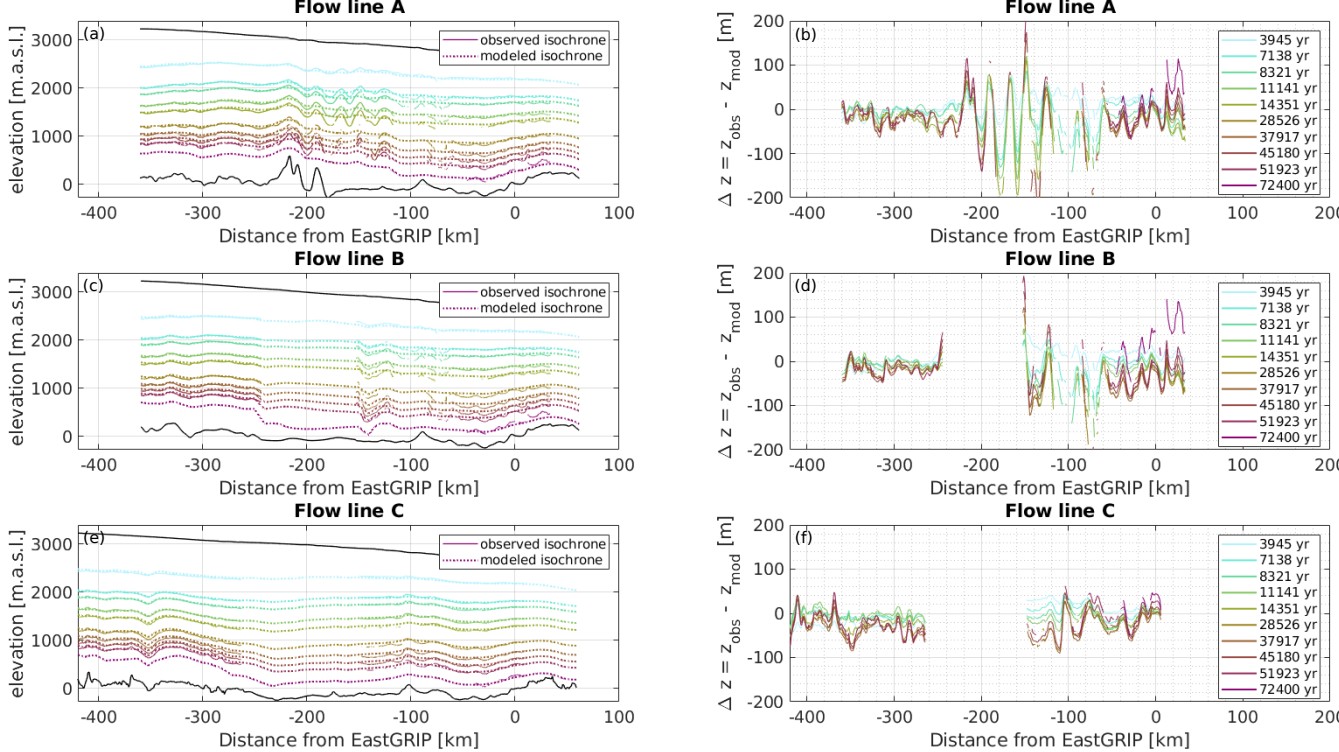

**Figure 5. (a, c, e)** Modelled and observed isochrones for profile A–C. The model fits the isochrones well in general but fails to reproduce strong layer undulations over short distances, leading to a larger misfit (panel **b, d, f**) where such undulations are present. A positive misfit indicates that the modelled isochrone depth is overestimated which happens in particular for the deepest isochrone towards the end of flow line A and B. As in Fig. 4 and Fig. 7, the color code represents the age of the corresponding isochrone.

point towards a good parameter resolution, while multiple maxima or large standard deviations indicate that several models are found to be equally likely. The parameter resolution is in general better in the beginning of the profiles, most clearly represented by the narrow distributions in the accumulation rate, basal melt rate and kink height at GRIP. Exponential distributions imply that a parameter reaches regularization boundaries. This is the case for the basal melt rates at GRIP, the kink height and basal sliding factor at EastGRIP, and the accumulation rate in flow lines B and C at EastGRIP. The climate parameter $c_w$ is found to be $0.10 \pm 0.005$ for all flow lines. The obtained value for parameter $c_c$ is $0.14 \pm 0.003$ for flow line A and B, and $0.16 \pm 0.004$ for flow line C respectively. The histograms of $c_w$ and $c_c$ can be found in the supplementary material, Fig. S3.

### 3.3 Ice origin and ice-flow history

From the modelled velocity field, we calculate particle trajectories backwards in time (Fig. 4) which give insight into the source location and flow history of ice found at a certain depth in the EastGRIP ice core, and allow us to determine the accumulation rate during its deposition (Fig. 7e). Due to the higher velocities in the ice stream, the ice source location in the upper 1,600 m

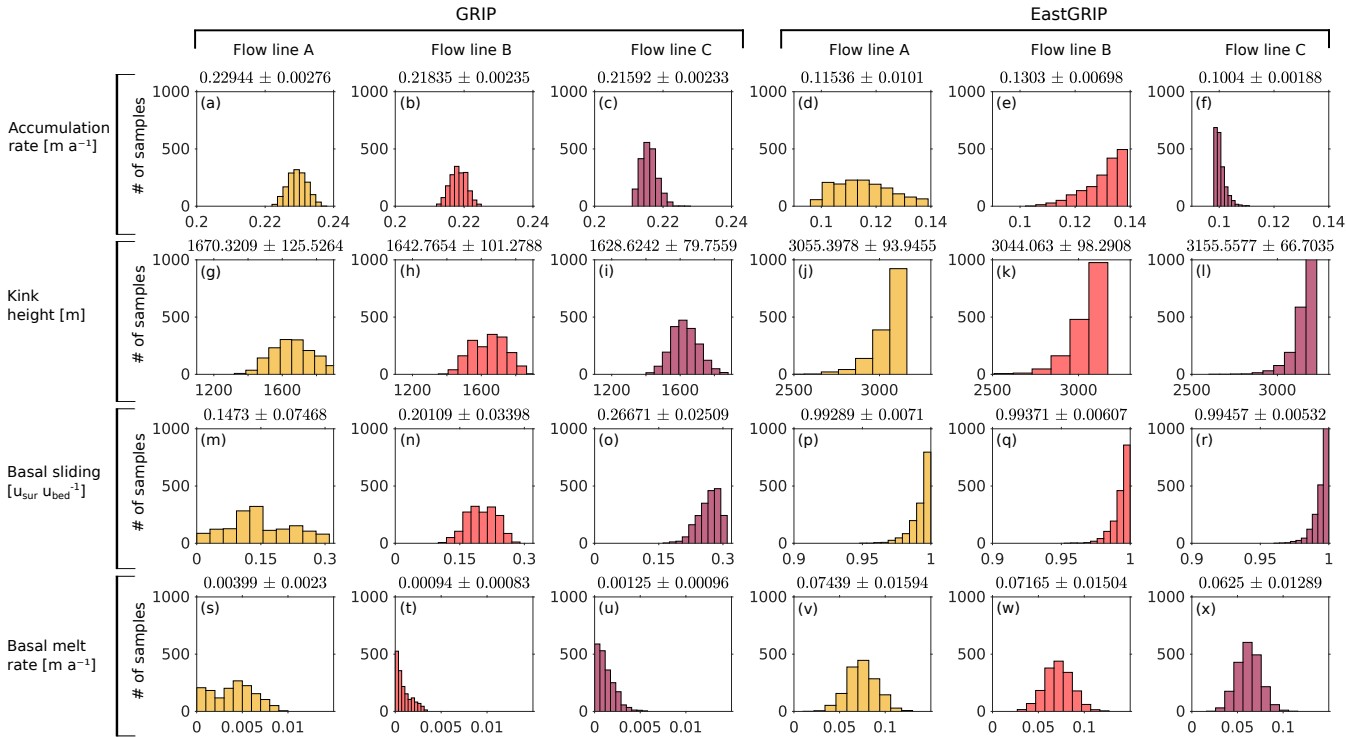

**Figure 6.** Histograms of the model parameters accumulation rate, basal melt rate, kink height and basal sliding at GRIP and EastGRIP for each flow line. The corresponding means and standard deviations are displayed on top of the histograms.

of the ice core lies further upstream for flow line C compared to flow line A and B. For deeper ice, this trend is reversed, as the velocity along flow line C drops below the velocity of line A and B (Fig. 7a). A similar effect manifests itself in the upstream elevation, where higher velocities along flow line C result in higher elevations in the upper part of the ice column, which is

compensated by a flatter topographic profile for ice deeper than 1,400 m (Fig. 7b).

From the model-inferred in situ accumulation rates, $\dot{a}_m$, and annual layer thicknesses, $\lambda_m$, we calculate the ice-core thinning function $\gamma$:

$$\gamma = \frac{\dot{a}_m - \lambda_m}{\dot{a}_m}. \tag{25}$$

The thinning function increases nearly linearly with depth in the Holocene and shows a considerable decrease in the Younger

Dryas and enhanced thinning in the Bølling–Allerød. In the glacial part of the ice core, the thinning function fluctuates between interstadials and stadials. The shift between the three lines results from the slightly different depth–age relationship and isochrone misfit obtained from the three profiles. We combine the thinning function with the annual layer thicknesses observed in the EastGRIP ice core, $\lambda_{obs}$, to estimate past accumulation rates $\dot{a}_{past}$:

$$\dot{a}_{past} = \frac{\lambda_{obs}}{1 - \gamma}. \tag{26}$$

**Table 5.** Essential quantities for upstream corrections for selected depths of the EastGRIP ice core. The upstream distance, elevation and past accumulation rates, $\dot{a}_{past}$, describe the characteristics of the source location and the conditions during ice deposition. $\dot{a}_{present}$ represents the corresponding present-day accumulation rates at the source location. All quantities are averages over the three flow lines and the uncertainties represent the maximum deviation from the mean.

| Depth | Age | Upstream distance | Elevation | Thinning function | $\dot{a}_{past}$ | $\dot{a}_{present}$ |
|---|---|---|---|---|---|---|
| [m] | [yr b2k] | [km] | [m.a.s.l.] | | [ma$^{-1}$] | [ma$^{-1}$] |
| 100 | 665 | $47 \pm 3$ | $2{,}752 \pm 10$ | $0.10 \pm 0.03$ | $0.12 \pm 0.004$ | $0.12 \pm 0.015$ |
| 200 | 1,553 | $74 \pm 2$ | $2{,}788 \pm 11$ | $0.19 \pm 0.08$ | $0.14 \pm 0.006$ | $0.14 \pm 0.005$ |
| 300 | 2,418 | $92 \pm 1$ | $2{,}837 \pm 14$ | $0.16 \pm 0.05$ | $0.13 \pm 0.005$ | $0.15 \pm 0.010$ |
| 400 | 3,322 | $105 \pm 7$ | $2{,}854 \pm 6$ | $0.21 \pm 0.02$ | $0.14 \pm 0.002$ | $0.14 \pm 0.031$ |
| 600 | 5,037 | $126 \pm 12$ | $2{,}892 \pm 14$ | $0.28 \pm 0.00$ | $0.16 \pm 0.001$ | $0.16 \pm 0.005$ |
| 800 | 6,805 | $146 \pm 14$ | $2{,}920 \pm 9$ | $0.35 \pm 0.03$ | $0.15 \pm 0.005$ | $0.16 \pm 0.003$ |
| 1,000 | 8,640 | $165 \pm 13$ | $2{,}944 \pm 15$ | $0.42 \pm 0.03$ | $0.16 \pm 0.004$ | $0.17 \pm 0.002$ |
| 1,200 | 11,015 | $183 \pm 12$ | $2{,}965 \pm 19$ | $0.41 \pm 0.06$ | $0.12 \pm 0.010$ | $0.17 \pm 0.005$ |
| 1,400 | 15,571 | $200 \pm 7$ | $2{,}993 \pm 7$ | $0.46 \pm 0.01$ | $0.05 \pm 0.001$ | $0.17 \pm 0.015$ |
| 1,600 | 23,382 | $217 \pm 3$ | $3{,}027 \pm 26$ | $0.52 \pm 0.08$ | $0.05 \pm 0.007$ | $0.18 \pm 0.023$ |
| 1,800 | 33,524 | $234 \pm 8$ | $3{,}054 \pm 40$ | $0.72 \pm 0.06$ | $0.11 \pm 0.028$ | $0.19 \pm 0.021$ |
| 2,000 | 43,107 | $252 \pm 14$ | $3{,}079 \pm 54$ | $0.73 \pm 0.07$ | $0.10 \pm 0.019$ | $0.19 \pm 0.022$ |
| 2,200 | 54,864 | $271 \pm 19$ | $3{,}108 \pm 73$ | $0.83 \pm 0.02$ | $0.07 \pm 0.006$ | $0.19 \pm 0.030$ |
| 2,400 | 75,980 | $293 \pm 18$ | $3{,}136 \pm 80$ | $0.83 \pm 0.11$ | $0.08 \pm 0.034$ | $0.19 \pm 0.023$ |
| 2,600 | 94,696 | $322 \pm 12$ | $3{,}171 \pm 67$ | $0.94 \pm 0.03$ | $0.18 \pm 0.044$ | $0.20 \pm 0.005$ |

We find that the accumulation rate at the deposition site increases from $\sim 0.12$ ma$^{-1}$ to a maximum of 0.249 ma$^{-1}$ for ice at a depth of 912 m, which was deposited approximately 7,800 years b2k. We note that the constant annual layer thicknesses observed in the upper 900 m of the EastGRIP ice core (Mojtabavi et al., 2020) coincides with the spatial pattern of increasing accumulation along the flow line with increasing upstream distance (Fig. 4c,g,k and Fig. 7d,e). Ice between 900 m and 1,400 m is characterized by the transition from the Holocene into the Last Glacial Period with decreased accumulation rates in the

Younger Dryas and a peak during the Bølling–Allerød (Fig. 7e). The accumulation rate at the deposition site for older ice varies between 0.02 ma$^{-1}$ during stadials and 0.196 ma$^{-1}$ during interstadials. The atmosphere in the glacial period was in general colder and dryer, and hence, accumulation rates were typically lower than today (Cuffey and Clow, 1997). However, due to the upstream flow effects, the ice from the interstadials could have been deposited under higher accumulation rates than are observed at the EastGRIP site today.

The variations in the past accumulation-rate between the three flow lines result from both, the varying along-flow accumulation pattern and different upstream distance of the source location. The spread between the three models provides important uncertainty estimates. The average deviation from the mean accumulation rates is 3.9 % in the Holocene and 20 % in the Last

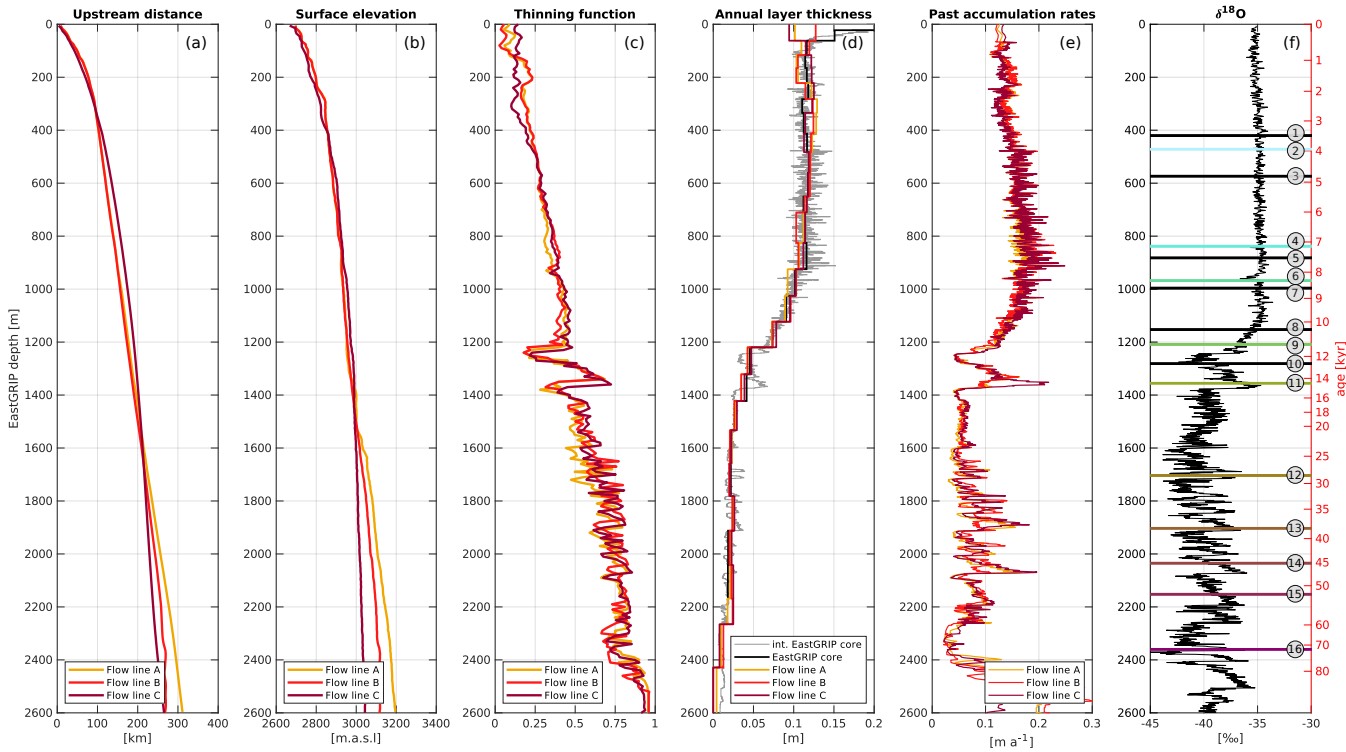

**Figure 7.** Modelled upstream distance **(a)** and surface elevation **(b)** of the source location for ice in the EastGRIP ice core. The thinning function **(c)** was calculated from the modelled accumulation rates and annual layer thicknesses **(d)** and was combined with the interpolated annual layer thicknesses observed in the ice core **(d)** to calculate past accumulation rates in high resolution **(e)**. The $\delta^{18}O$ curve from NorthGRIP **(f)** was scaled to the EastGRIP depths to put the results into a climatic context. The depths of the traced isochrones from Table 3 are displayed with the same color index as in Fig. 4 and Fig. 5 and labeled with the corresponding layer number.

Glacial Period. The largest spread between the three flow lines is 68 % observed at a depth of 2,411 m. We remark that, due to missing direct information on the annual layer thicknesses, accumulation rates below the current borehole depth of 2,122.45 m are based on tentative estimates and must be treated accordingly.

## 4 Discussion

### 4.1 Isochrone deformation and ice-flow parameters

Deformation of IRHs occurs as a consequence of bed topography (Robin and Millar, 1982; Jacobel et al., 1993), spatial variations in basal conditions (Weertman, 1976; Whillans, 1976; Whillans and Johnsen, 1983; Catania et al., 2010; Christianson et al., 2013; Leysinger Vieli et al., 2018; Wolovick et al., 2014), spatially varying accumulation rates and corresponding changes in ice-flow geometry (Dansgaard and Johnsen, 1969; Weertman, 1976; Whillans, 1976; Whillans and Johnsen, 1983), and as

a consequence of convergent ice flow and ice-stream activity (Bons et al., 2016). Areas of enhanced basal melt rates similarly drag down all the layers above, while variations in accumulation rate, kink height and basal sliding lead to depth-dependent deformation of the isochrones (Keisling et al., 2014).

The accumulation rates of $\sim$0.21–0.23 $\mathrm{ma}^{-1}$ at GRIP and $\sim$0.1–0.13 $\mathrm{ma}^{-1}$ at EastGRIP obtained in this study agree with field observations (Dahl-Jensen et al., 1993; Vallelonga et al., 2014), and the low standard deviations point towards a robust solution. In profiles A and B we observe $\sim$20 % lower accumulation rates inside the ice stream than outside. This agrees to some extent with Riverman et al. (2019), who found 20 % higher accumulation rates in the shear margins compared to the surrounding, although our observations are not confined to the shear margins only. Regularization on the accumulation rate

was necessary in our model to avoid unrealistic strong fluctuations along the flow lines.

The bed topography and bed lubrication have a considerable effect on ice-flow parameters. Flow over bed undulations affect the elevation of internal layers due to variations in the longitudinal stresses within the ice (Hvidberg et al., 1997) and is often reflected in the surface topography (Cuffey and Paterson, 2010). If the bed is 'sticky', i.e. the basal sliding is small, the ice is compressed along the flow direction while vertically extended (Weertman, 1976), and IRHs are pushed upwards. At a slippery

bed, the opposite is the case, resulting in along-flow extension of IRHs which leads to thinning and thus decreasing distance between the IRHs. Keisling et al. (2014) argued that major fold trains existing independently of bed undulations can be explained by variations in the basal sliding conditions. This is, for instance, observed across shear margins, where local, steady state folds are formed as a response to the basal conditions (Keisling et al., 2014; Holschuh et al., 2014). In flow line A, we observe similar 'fold-trains' on a larger scale downstream of a substantial bed undulation (100–200 $\mathrm{km}$ upstream of EastGRIP), and the

resulting high basal melt rates and sliding fraction and the low kink heights drag the layers down in the attempt of matching the observed synclines. These strongly deformed isochrones predominantly appear in parts of the flow lines which deviate from the observed surface velocity direction by more than 15 degrees. We thus argue that they are out-of-the-plane effects and that the isochrones along the ice-flow direction are less strongly deformed. Accordingly, it is questionable that the ice in the EastGRIP ice core has experienced such deformation and that the high local basal melt rates are trustworthy. The fact that these folds

are not reproduced very well by the model does therefore not put any constraints on the usefulness of our results for upstream corrections.

The NEGIS differs from other ice streams in Greenland and Antarctica through the lack of clear lateral topographic constraints and high ice-flow velocities reaching exceptionally far inland. The positioning of the shear margins of the NEGIS are most likely strongly interconnected to the subglacial water system and the substrate and morphology of the bed (Christianson

et al., 2014; Franke et al., 2021a). The vast amount of ice mass is added to the NEGIS by entering the ice stream through the shear margins (Franke et al., 2021a), resulting in a compressional stress regime perpendicular to the ice stream. The sudden increase in the kink height at around 60 $\mathrm{km}$ upstream of EastGRIP pushes the isochrones upwards, similar to the effect of lateral compression. The distribution of available melt water and a soft, deformable bed facilitate sliding and thus, ice-flow acceleration at the NEGIS onset (Christianson et al., 2014). Evidence of a locally enhanced geothermal heat flux and basal

ice at the melting point has been presented by e.g. Fahnestock et al. (2001) and MacGregor et al. (2016), and bed lubrication through melt-water production seems to be one of the driving mechanisms for rapid ice flow in the onset region of the NEGIS

(Smith-Johnsen et al., 2020b). Our results support these previous findings in the following way: (1) Kink heights close to the bed in large segments along the flow profiles imply that most shear deformation is happening in the lower part of the ice column or at the ice–bed interface. The increased kink height towards the ends of the profiles can be attributed to the compressional stress regime associated with the addition of ice through the shear margins. (2) Basal melt rates of 0.01 $\mathrm{ma}^{-1}$ or higher inside the NEGIS suggest that the basal ice temperatures along the flow lines are at the pressure melting point and enough energy is available to produce melt water leading to substantial bed lubrication. (3) Basal sliding is present in most segments of the flow lines, suggesting the presence of melt water or deformation of a soft bed. It increases considerably along the flow lines and significantly contributes to the surface velocity at EastGRIP.

While it is commonly accepted that the NEGIS is initiated by a locally enhanced geothermal heat flux (e.g. Fahnestock et al., 2001; Alley et al., 2019), the magnitude thereof and the resulting hydrological conditions of the bed are still highly debated. Previous studies using simple strain-rate models in combination with Holocene radar stratigraphy indicate basal melt rates of 0.1 $\mathrm{ma}^{-1}$ or higher in the vicinity of EastGRIP (Fahnestock et al., 2001; Keisling et al., 2014; MacGregor et al., 2016). However, the accuracy of these findings is limited since the local layer approximation (Waddington et al., 2007) is not valid in the surrounding of the NEGIS (Keisling et al., 2014; MacGregor et al., 2016). Remarkably high basal melt rates of 0.16–0.22 $\mathrm{ma}^{-1}$ are also suggested by a recent study (Zeising and Humbert, 2021) using an autonomous phase-sensitive radio-echo sounder (ApRES) at EastGRIP. Melt rates in these order of magnitudes would either require an unusual high geothermal heat flux exceeding the continental background (Fahnestock et al., 2001; Bons et al., 2021) or an additional heat source (Zeising and Humbert, 2021). Alley et al. (2019) discussed the interactions between the GrIS and the geothermal anomaly, presumably caused by the passage of Greenland over the Iceland hot spot (Lawver and Müller, 1994), and hypothesized that an exceptionally unsteady and inhomogeneous geothermal heat flux underneath northeast Greenland could arise through perturbations of the mantle stress regime caused by ice-sheet fluctuations.

Our results indicate basal melt rates at EastGRIP between 0.05 and 0.1 $\mathrm{ma}^{-1}$ but higher values of up to 0.2 $\mathrm{ma}^{-1}$ are obtained further downstream. However, the depth of the oldest modelled isochrone tends to be overestimated in this part of the flow lines (Fig. 5), indicating that the basal melt rate is overestimated. Ice-flow parameters at a certain location affect the isochrones directly above and further downstream and since EastGRIP is near the end of the radar lines, the information constraining the isochrone depths is limited, leading to overall lower parameter resolution than further upstream.

## 4.2 EastGRIP source location and upstream effects

The source region of ice in the EastGRIP ice core extends over more than 300 $\mathrm{km}$ upstream. Holocene ice characterizes the upper 1,244 $\mathrm{m}$ of the ice core and has been advected up to 197 $\mathrm{km}$. The climatic conditions during the last 8 kyr remained nearly constant with similar accumulation rates as today. However, due to increasing precipitation towards the central ice divide, ice from the past 8 kyr was deposited under increasingly higher accumulation rates with increasing age (Table 5). Our results indicate that this upstream effect happens to compensate for the vertical layer thinning and results in the constant annual layer thicknesses observed in the upper 900 $\mathrm{m}$ of the EastGRIP ice core (Mojtabavi et al., 2020). One possible conclusion of this peculiar observation is that snow depositions must have been advected from far enough upstream to allow the compensation

of vertical thinning by increased accumulation rates in the source location. This gives reason to the hypothesis that ice-flow velocities in the past 8 kyr must have been similarly fast as today, and that, therefore, the NEGIS has likely been active during this time. However, we believe that RES images and estimates of present-day accumulation rates along the EastGRIP flow line are necessary to evaluate this hypothesis further.

Between 8 ka b2k and the beginning of the Holocene, accumulation rates decreased at the deposition site due to progressively colder and dryer climatic conditions (Cuffey and Clow, 1997) as we go further back in time and transition into the GS-1. The most recent Glacial Period extends from 119,140 to 11,703 years b2k (Walker et al., 2009) and is characterized by Dansgaard–Oeschger events, abrupt transitions between cold stadial and relatively mild interstadial periods (Dansgaard et al., 1982; Johnsen et al., 1992) causing the oscillations in the accumulation rates. Ice from the Last Glacial Period was deposited

between 197 and 332 km upstream from EastGRIP. The basal ice at EastGRIP could be more than 100 ka old which, according to our models, has been deposited within about 50 km from the ice divide under conditions similar to those at NorthGRIP and GRIP.

Ice that is entering the NEGIS must somehow penetrate the shear margin, which is an important characteristic of ice flow in ice streams and might have left an imprint on the crystal fabric and texture of ice extracted at EastGRIP. Our modelling results

along flow line A and B indicate that ice below 239 m in the EastGRIP ice core has passed the shear margin 82 km from EastGRIP around 1.8 ka b2k. Slightly enhanced annual layer thicknesses observed in the ice core at a depth of 230 m (Fig. 3) seem unrelated to short-term warmer and wetter climate and might thus be an effect of enhanced accumulation across the shear margin, supporting our results.

Our results show surface elevations at the deposition site which are up to 500 m higher than EastGRIP at the corresponding

time. Assuming a normal thermal and pressure gradient, this implies that ice was deposited under up to $\sim$3.25°C colder temperatures and up to 45 hPa lower pressure than conditions found at the borehole location at the time of deposition.

## 4.3    Limitations

The most relevant limitation of this study arises from lacking radar data parallel to the flow field in the upstream area of EastGRIP. The approximated flow lines deviate from the present-day surface flow field in some parts by more than 15 degrees,

which presumably introduces out-of-the-plane effects. Data gaps encumbered isochrone tracing and restricted the Monte-Carlo method due to missing information in those areas. The correlated parameters in the Dansgaard-Johnsen model lead to a vast amount of possible solutions, and the fact that the observed data can be reproduced by our model does not prove the validity of the assumed parameters and the physical interpretation thereof. This becomes for instance evident at the GRIP ice-core site, where our results indicate basal sliding of up to 30 %, while the drilling project showed that the bed is frozen at the ice-sheet

summit (Dahl-Jensen et al., 1998). The apparant basal sliding might thus represent deformation within a soft bed material rather than actual sliding of the ice over the bed. The spatial and formal resolution of the obtained model parameters is limited, in particular towards the end of the profiles due to limited constraining information further downstream.

By introducing the parameter $\alpha$, our model accounts for lateral compression and extension on a first degree order, but does not capture the full complexity of the flow field across the shear margins. While these play an essential role in the ice-flow

dynamics of the NEGIS (Holschuh et al., 2019) and are likely to have left an imprint on the ice found in the EastGRIP ice core, the full simulation of the flow field is not attempted for the purpose of upstream corrections. This would require more complex, 3D numerical ice-flow models which are computationally more expensive and thus not suitable for the Monte-Carlo method applied here. Moreover, due to the lack of constraining radar data the information gain in terms of the source characteristics and upstream effects of such a 3D model would be modest.

The elevation of the source location was determined solely from the present-day ice-sheet surface elevation and did not take into account past fluctuations in the ice-sheet thickness. In general, surface elevation changes are relatively minor in the interior areas of central Greenland (Marshall and Cuffey, 2000; Letréguilly et al., 1991). Yet, Vinther et al. (2009) found that the GRIP elevation might have been up to 200 m higher during the early Holocene than today. We did not take into account changes in the ice thickness due to the large uncertainties which would be introduced, particularly in the Last Glacial Period. Our estimates 505 on the surface elevation of the source location must thus not be interpreted as absolute values but rather as relative changes with respect to the surface elevation of the EastGRIP site at the corresponding time.

Lacking data and a general understanding of ice-sheet flow far back in time put up additional constraints, and due to the relatively recent discovery of the NEGIS (Fahnestock et al., 1993), little is known about its evolution in the past. Observations of surface elevation and ice-flow velocities imply that the downstream end of the NEGIS has entered a state of dynamic thinning 510 after at least 25 years of stability (Khan et al., 2014). However, it is not clear for how long the NEGIS has been active and how its catchment geometry changed over time. The assumption of a constant flow field throughout the past 100 kyr is thus the best currently available, but potentially inaccurate, estimate of the past flow regime.

Our results do not give clear evidence on which of the flow lines gives the best results for upstream corrections. Since the present-day EastGRIP flow line is likely located somewhere between flow line A and C, our results can be interpreted as the 515 outer boundaries and we consider the average over the three flow lines the best estimate for the upstream flow characteristics with the corresponding model spread as uncertainties.

## 5    Conclusions

We traced isochrones in RES images along three approximated EastGRIP flow lines connecting the EastGRIP and GRIP drill sites. A two-dimensional Dansgaard–Johnsen model was used to simulate the propagation of isochrones along these flow lines. 520 The simplicity of the model allowed us to invert for the ice-flow parameters accumulation rate, basal melt rate, kink height and basal sliding fraction, which give limited but helpful insight into basal properties and ice-flow dynamics and can be used to constrain large-scale ice-sheet models.

On the basis of our modelled two-dimensional velocity field, we calculated particle trajectories backwards in time to determine the deposition site of ice found in the EastGRIP ice core. We present estimates of the upstream distance, surface elevation 525 and accumulation rate at the time and location of ice deposition. This is valuable and necessary information for interpreting ice-core measurements, and to separate past climate variability from non-local imprints introduced by upstream effects. Our studies show that spatially increasing accumulation rates with increasing upstream distance along the flow line are mainly

responsible for the constant annual layer thicknesses observed for the last 8 kyr in the EastGRIP ice core.

The lack of radar data along the EastGRIP flow line is the biggest limitation of this study. None of the three simulated flow lines accurately represents the present-day flow field but can be regarded as upper and lower limits framing the upstream effects. The acquisition of further radar data along NEGIS flow lines in the future would thus provide more accurate and valuable insights into the flow history of the EastGRIP ice and the NEGIS.

*Data availability.* The CReSIS radio-echo-sounding images used for isochrone tracing are publicly available on https://data.cresis.ku.edu/. The RES data recorded by AWI will be available by Jansen et al. (2020) and described by (Franke et al., 2021b). The extended EastGRIP time scale, our derived and approximated flow lines and an extended version of Table 5 will be available on https://www.iceandclimate.nbi.ku.dk/data/ and in the supplementary material to this paper.

## Notation

| | |
|---:|:---|
| $z$ | height above bed |
| $x, y$ | direction parallel, perpendicular to the flow profile |
| $\tilde{z}_t, \tilde{z}_p, \tilde{z}_{rr}$ | total, picking related, radar related isochrone depth uncertainty |
| $k$ | radar window widening factor |
| $c$ | speed of light |
| $B$ | radar band width |
| $\tilde{a}_t, \tilde{a}_p, \tilde{a}_{rr}, \tilde{a}_c$ | total, picking related, radar related, ice-core related age uncertainty |
| $a_c$ | ice-core age |
| $u_\parallel(z), u_{\parallel,sur}$ | flow-line-parallel velocity at depth, at the surface |
| $u_\perp(z), u_{\perp,sur}$ | flow-line-perpendicular velocity at depth, at the surface |
| $\hat{\mathbf{u}}_{sur}$ | normalized surface velocity along the flow line |
| $\omega(z), \omega_{sur}, \omega_{bed}$ | vertical velocity at depth, at the surface, at the bed |
| $f_{bed}, f_{bed,0}$ | basal sliding fraction, initial basal sliding fraction |
| $h, h_0$ | kink height, initial kink height |
| $H$ | ice thickness |
| $\alpha$ | sum of horizontal strain rates $\dot{\varepsilon}_{xx} + \dot{\varepsilon}_{yy}$ |
| $E_{bed}$ | bed elevation above sea level |
| $E_{sur}$ | surface elevation above sea level |
| $\dot{a}, \dot{a}_m, \dot{a}_c, \dot{a}_{past}, \dot{a}_{present}$ | positive accumulation rate, Monte-Carlo inferred, ice-core inferred, past, present |
| $\dot{b}$ | positive basal melt rate |
| $e_{1,2,3}$ | scaling factors for initial model parameters |
| $\xi(t)$ | climatic scaling factor |
| $t$ | time b2k |
| $c_c, c_w$ | sensitivity of accumulation rates in cold stadial, warm interstadial periods |
| $\partial^{18}O_c, \partial^{18}O_w$ | typical isotope values for cold stadial, warm interstadial periods |
| $\mathbf{d}, \mathbf{d}_{obs}, \mathbf{d}_{model}$ | data space, observed, modelled data |
| $\mathbf{m}, \mathbf{m}_{old}, \mathbf{m}_{new}$ | model space, old, new model |
| $\sigma_z, \sigma_\alpha$ | data uncertainty on isochrone depth, $\alpha_{sur}$ |
| $P_{accept}$ | acceptance probability |
| $L$ | likelihood function |
| $S$ | misfit function |
| $\lambda, \lambda_m, \lambda_{obs}$ | annual layer thickness, modelled, observed |
| $\gamma$ | ice-core thinning function |
| $\dot{\varepsilon}_{xx}, \dot{\varepsilon}_{yy}, \dot{\varepsilon}_{xy}$ | horizontal strain rates |

## Appendix A: Derivation of Eq. (14)

The vertical velocity in the upper part of the ice column is given by

$$w(z) = w_{sur} + \alpha_{sur}(H - z) = \frac{dz}{dt}, \qquad z \in [h, H]. \tag{A1}$$

Assuming that the surface slope is close to zero, $\frac{\partial E_{sur}}{\partial x} \simeq 0$, integration of Eq. (A1) with separation of variables results in:

$$\int_{z}^{H} \frac{1}{-\dot{a} + \alpha_{sur}(H - z)} dz = \int_{t}^{0} 1 \, dt \tag{A2}$$

$$\left[ -\frac{1}{\alpha_{sur}} \ln(\alpha_{sur}(H - z) - \dot{a}) \right]_{z}^{H} = [t]_{t}^{0} \tag{A3}$$

$$-\frac{1}{\alpha_{sur}} (\ln(-\dot{a}) - \ln(\alpha_{sur}(H - z) - \dot{a})) = -t \tag{A4}$$

$$(H - z) = \frac{\dot{a}}{\alpha_{sur}} (1 - e^{-\alpha_{sur}t}) \tag{A5}$$

## Appendix B: Definition of the misfit function S(m)

The function $S(\mathbf{m})$ is defined as:

$$S(\mathbf{m}) = \frac{1}{2} \left( \frac{1}{10} \sum_{l=1}^{10} \left( \frac{1}{n_z} \sum_{n=1}^{n_z} \mathbf{M}_z^2 \right) + \frac{1}{n_\alpha} \sum_{n=1}^{n_\alpha} \mathbf{M}_\alpha^2 \right) * 1000, \tag{B1}$$

where $l$ runs through the 10 layers, $n_z$ and $n_\alpha$ are the number of observed isochrone depths and $\alpha_{sur}$ along the flow lines, and

$$\mathbf{M}_z = \frac{\mathbf{d}_{mod,z} - \mathbf{d}_{obs,z}}{\sigma_z} \quad \text{and} \quad \mathbf{M}_\alpha = \frac{\mathbf{d}_{mod,\alpha} - \mathbf{d}_{obs,\alpha}}{\sigma_\alpha}. \tag{B2}$$

The matrix $\mathbf{M}_z$ describes the misfit between modelled ($\mathbf{d}_{mod,z}$) and observed ($\mathbf{d}_{obs,z}$) isochrones, and the vector $\mathbf{M}_\alpha$ is the misfit between modelled ($\mathbf{d}_{mod,\alpha}$) and observed ($\mathbf{d}_{obs,\alpha}$) $\alpha_{sur}$. The data uncertainty $\sigma_z$ is the maximum depth uncertainty of 13 m, and the uncertainty on $\alpha_{sur}$ ($\sigma_\alpha$) is assumed to be 10 % of the maximum observed $\alpha_{sur}$. The factor 1000 is a tuning parameter to ensure the acceptance ratio remains between 25 % and 75 %.

*Author contributions.* DDJ and TAG designed and carried out the study. AG developed the code used for isochrone tracing. AG and CSH derived the EastGRIP flow lines. DJ was co-investigator for the AWI radar survey and acquired the data in the field. SF processed the radar data obtained during the EGRIP-NOR-2018 AWI flight campaign. SOR, GS and TAG synchronized the EastGRIP ice core with NorthGRIP and NEEM and extended the time scale to the current drill depth. TAG prepared the manuscript with the contribution of all co-authors.

*Competing interests.* The authors declare that they have no conflict of interest.

*Acknowledgements.* This work was supported by the Villum Investigator Project IceFlow (NR. 16572). EastGRIP is directed and organized by the Centre for Ice and Climate at the Niels Bohr Institute, University of Copenhagen. It is supported by funding agencies and institutions in Denmark (A. P. Møller Foundation, University of Copenhagen), USA (US National Science Foundation, Office of Polar Programs), Germany (Alfred Wegener Institute, Helmholtz Centre for Polar and Marine Research), Japan (National Institute of Polar Research and Arctic Challenge for Sustainability), Norway (University of Bergen and Trond Mohn Foundation), Switzerland (Swiss National Science Foundation), France (French Polar Institute Paul-Emile Victor, Institute for Geosciences and Environmental research), Canada (University of Manitoba) and China (Chinese Academy of Sciences and Beijing Normal University). We acknowledge the use of data and data products from CReSIS generated with support from the University of Kansas, NASA Operation IceBridge grant NNX16AH54G, NSF grants ACI-1443054, OPP-1739003, and IIS-1838230, Lilly Endowment Incorporated, and Indiana METACyt Initiative. We also acknowledge the use of the CReSIS toolbox from CReSIS generated with support from the University of Kansas, NASA Operation IceBridge grant NNX16AH54G, and NSF grants ACI-1443054, OPP-1739003, and IIS-1838230. We thank the crew of the research aircraft Polar 6 and system Engineer Lukas Kandora for their work during the AWI flight campaign 2018 and express our gratitude to John Paden and Tobias Binder, who helped with the data acquisition during the AWI survey. Sune Olander Rasmussen and Giulia Sinnl gratefully acknowledge the Carlsberg Foundation for supporting the project ChronoClimate. This research was enabled in part by computing facilities and support provided by WestGrid (www.westgrid.ca) and Compute Canada Calcul Canada (www.computecanada.ca). The scientific colour maps 'hawaii' and 'roma' (Crameri, 2020) are used in this study to prevent visual distortion of the data and exclusion of readers with colourvision deficiencies (Crameri et al., 2020).

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
