# Peer review of "Upstream flow effects revealed in the EastGRIP ice core using a Monte Carlo inversion of a two-dimensional ice-flow model"

_The Cryosphere, 2021_

## Referee Comment (RC1)

**Review of the article entitled "Upstream flow effects revealed in the EastGRIP ice core using a Monte Carlo inversion of a two-dimensional ice-flow model"**

**1  General comments**

This study investigates the ice flow on the upstream part of the NorthEast Greenland Ice Stream (NEGIS) in order to provide information relative to ice origin to the East Greenland Ice-Core Project (EastGRIP). The authors use a 2 dimensional, flowline Dansgaard-Johnsen model to simulate the ice flow upstream of the EastGRIP drilling location. The simulations are performed along three lines that roughly approximate the ice stream flow but have the advantage of being along measured radar profiles. The use of the radar profile allows to constrain the parameters of the model through a Monte Carlo inversion that minimises the misfit between observed and modelled isochrones. The model finally allow to compute trajectories of ice particles backwards in time from the EastGRIP borehole to define the location, accumulation rate and altitude at the time of deposition. These information will help to further analyse the results from the ice core and already show that the constant annual layer thickness observed in the first 900 meters of the EastGRIP ice core is mainly due to the increase of the accumulation rate as the ice source gets closer to the ice sheet divide.

In general, this paper is well written and nicely illustrated. The conclusions regarding the localisation of the deposition point in the past are well documented and seems to be robust given the methods that are used. It seems however that some key information are missing on the parameter selection for the Dansgaard-Johnsen model and more particularly regarding the regularisation of the data and discussion on their spread. I am also puzzled by some of the differences that can be observed on the overlapping flowline and this should be discussed further.

1. It is stated that a threshold is used to regularise the maximum deviation of the parameters and allow to keep those parameters in a physically feasible range. It is however unclear what this range is and how it was defined. Form Figure 5 it seems that the threshold is exceeded for the basal melt rate for flowlines A and B at

EastGRIP. However, the values presented here are lower than the maximum values for basal melt that have been recently presented at this location. It feels that for this parameter the threshold might need to be increased to allow the basal melt rate to increase. In any case, more information should be given on how those threshold and the "physically feasible range" have been selected. On the same figure 5 we can also see a very large standard deviation on the kink height at EastGRIP, together with a large spread of the mean value across the different flowlines. It would be nice to get more insight into the effect of this variation and the possible implication of this large spread if ever a parameter far from the actual value should be used.

2. Lines A and B share the same line close to EastGRIP, However in Figure 3 the velocity and stresses in the region where the line is common are different. From my understanding the velocities have been extracted from the MEaSUREs dataset along the radar profile so I don't understand why those plots are different. On Figure 4 we also see that the modelled isochrone on the region where A and B are on the same radar line show significantly different results. I expect that this arises from the different parameter selected through the Monte Carlo approach for the model but it would be interesting to discuss this point further and potentially identify a "better" set of parameter from these portions of flowline.

3. As a general point I think that the limitations of the model should be developed a bit more. The authors state that the lack of radar data is the major limitation of this study but perhaps applying a more evolve 3D model would allow to overcome this issue. I understand that there is a trade off here between using a simple model that allows the Monte Carlo approach and using a highly under constrained more physically accurate models but the limitation inherent to this choice should be more clearly pointed out.

**2   Specific comments**

Bellow is a list of more specific and technical comments throughout the manuscript given with line numbers:

- Line 5: It is stated here that the location inside an ice stream introduces non climatic bias. But isn't that redundant with the high ice-flow velocities, or is it meant that specific processes linked to the ice stream such as shear margin are causing some of those biases?

- Line 8: It might be worth noting here that the selected flowline emanate from the Greenland Ice Sheet summit but that the site as also specific interest due to the presence of the GRIP ice core.

- Line 9: the RES abbreviation is not re-used in the abstract and could be dropped.

- Line 10: As the model is solved along a flowline, wouldn't the source be a point rather than an area.

- Line 26: I think that the statement relative to the modelling of NEGIS is not completely true and that more recent modelling work like the ones of Beyer et al. (2018); Smith-Johnsen et al. (2020) should be discussed here.

- Line 27: Some results from the EastGRIP ice core have started to appear but it might be more fair to use the future tense here as much more results are to be produced.

- Line 37: I am not sure what is referred to here when using the term "lateral flow", is that to contrast with vertical flow? If that is the case perhaps "horizontal flow" would fit better?

- Line 39: I would prefer "spatial variability of the precipitation".

- Line 52: The fact that the model is 2D vertical should be mentioned here.

- Line 63: The sentence starting on this line is unclear, perhaps "as a consequence of" should be dropped.

- Line 67: The deviation of the different flowline does not seem that extreme to me, particularly if they are compared to the spread of the radar lines that are used.

- Figure S1: On this figure I am missing a scale more convenient than the one on the map border which would help to judge distances better on the map.

- Figure 1: As for Figure S1, Figure 1 would benefit from a more convenient scale legend.

- Line 79: "the NEGIS trunk"

- Line 80: Line C also presents a quite substantial data gap and that should be noted here.

- Line 84: "centre" should be capitalised.

- Line 109: "Greenland Stadial 2 (GS-2)"

- Figure 2: It would be nice to show the different Greenland Stadials on the age axis.

- Line 138: This description of the dating process is not the clearest. I am not sure of what the 250m represent, is that following the radar line up and downstream to smooth out any local bump in the IRH? This whole sentence should probably be rephrased.

- Figure 3: On panel b the surface velocity legend is missing

- Figure 4: I think that "very well" to describe the fit of the isochrones is an over-statement. To my eye it seems that there is a bias with the modelled isochrones being slightly higher up in the ice column than the observed one.

- Figure 5: I suppose that the basal sliding is expressed as a fraction of the surface velocity. That should be stated in the Figure or in its caption. On panel (i) and (j) the exponent on the x axis is confusingly placed.

- Figure 6: There is a legend missing in panel (d).

- Line 295: It is not sure to me what "local accumulation" means in this context. From the rest of the sentence I expect that it is the accumulation at the deposition site but somehow "local" here make it unclear.

- Line 300: The sentence starting on this line is not completely clear. If I refer to the present-day accumulation stated above ($0.12$ ma$^{-1}$) the ($0.14$ma$^{-1}$) given here for interstadial is actually higher then present day.

- Line 377: The sentence starting on this line should be modified. Zeising and Humbert (2021) actually state in the last part of their paper that "We are aware that these melt rates require an extremely large amount of heat that we suggest to arise from the subglacial water system and the geothermal heat flux." and that they are able to close their energy budget with a more reasonable geothermal heat flux around $0.25$ Wm$^{-2}$.

- Line 388: "Propagate" might not be the good term here.

- Line 408:It would be nice here to have more information on the reason why this simulation is not attempted. Is it just due to the fact that the constraints on the model would be lacking, that it is not warranted for the specific goal of estimating the source location of the ice or for other reasons.

**References**

Beyer, S., Kleiner, T., Aizinger, V., Rückamp, M., and Humbert, A. (2018). A confined–unconfined aquifer model for subglacial hydrology and its application to the northeast greenland ice stream. *The Cryosphere*, 12(12):3931–3947.

Smith-Johnsen, S., de Fleurian, B., Schlegel, N., Seroussi, H., and Nisancioglu, K. (2020). Exceptionally high heat flux needed to sustain the northeast greenland ice stream. *The Cryosphere*, 14(3):841–854.

Zeising, O. and Humbert, A. (2021). Indication of high basal melting at eastgrip drill site on the northeast greenland ice stream. *The Cryosphere Discussions*, 2021:1–15.

---

## Author Comment (AC1)

**Authors response on Referee Comment tc-2021-63-RC1**

May 21, 2021

**1 General comments**

1. It is stated that a threshold is used to regularise the maximum deviation of the parameters and allow to keep those parameters in a physically feasible range. It is however unclear what this range is and how it was defined. From Figure 5 it seems that the threshold is exceeded for the basal melt rate for flowlines A and B at EastGRIP. However, the values presented here are lower than the maximum values for basal melt that have been recently presented at this location. It feels that for this parameter the threshold might need to be increased to allow the basal melt rate to increase. In any case, more information should be given on how those threshold and the "physically feasible range" have been selected. On the same figure 5 we can also see a very large standard deviation on the kink height at EastGRIP, together with a large spread of the mean value across the different flowlines. It would be nice to get more insight into the effect of this variation and the possible implication of this large spread if ever a parameter far from the actual value should be used.

   **Author response:** We agree that this part has not been addressed enough in the current version of the manuscript and we will include a more detailed explanation on the parameter regularisation in the revised version. The regularisation was done as follows:

   - The kink height is allowed to vary within the ice thickness, i.e. $0 < h < H$. Anything outside this range would physically not make sense. We did not regulate this parameter further because we do not have any prior knowledge on the vertical velocity distribution along the flow line.

   - The accumulation rate was set to vary within $\pm 2$ cm from the initial accumulation rate, i.e. $\lambda_{H,0} - 0.02 < \lambda_H < \lambda_{H,0} + 0.02$. The initial accumulation rate obtained from Eq. (12) fits quite well with field observations at GRIP and EastGRIP. Nevertheless, small deviations can be expected because the local layer approximation is not valid in the study area. We estimate that this deviation should be within $\pm 2$ cm from the initial guess.

   - The basal sliding was allowed to vary $\pm 15$ % from the initial parameterisation, i.e. $f_{B,0} - 0.15 < f_B < f_{B,0} + 0.15$, resulting in a maximum sliding fraction of 95 % in the vicinity of EastGRIP.

   - The threshold for the basal melt rate was set to 0.03 m a$^{-1}$, i.e. $\lambda_{B,0} - 0.03 < \lambda_B < \lambda_{B,0} + 0.03$ which allows a maximum melt rate of 0.06 m a$^{-1}$ at EastGRIP. The choice

of this upper limit is based on our preliminary modeling results which showed that much higher basal melt rates lead to erosion of the deepest observed isochrones.

Basal melt rates suggested by previous studies were obtained from the Holocene stratigraphy only (e.g. Fahnestock et al., 2001; Keisling et al., 2014; MacGregor et al., 2016) or represent present-day estimates (Zeising and Humbert, 2021), while in our model the basal melt rates are averages over the entire modeled time period (∼50ka). However, we agree that the upper limits of the basal melt rates are reached in flow line A and B and that these thresholds should be loosened for more accurate sampling of the posterior probability. We will address this in the revised version of the manuscript by running additional models with more loose parameter regularisation.

Any parameter perturbation at a certain point along the flow line affects the isochrones at that location and further downstream, i.e. the model parameters are not uncorrelated. A drag-down of the isochrones due to enhanced accumulation, for instance, can be compensated by enhanced kink height, low basal melt rate or a decreased basal sliding fraction. Because the kink height is the least regulated parameter it can fluctuate a lot and compensate for effects caused by other parameters. The parameter distribution at EastGRIP thus depends a lot on the parameters in the upstream area and a deviation between the flow lines can be expected even if the observed isochrones are identical in the downstream area. We will add a few sentences in the revised version of the manuscript to discuss the parameter correlation and their effects on our results.

2. Lines A and B share the same line close to EastGRIP, However in Figure 3 the velocity and stresses in the region where the line is common are different. From my understanding the velocities have been extracted from the MEaSUREs dataset along the radar profile so I don't understand why those plots are different. On Figure 4 we also see that the modelled isochrone on the region where A and B are on the same radar line show significantly different results. I expect that this arises from the different parameter selected through the Monte Carlo approach for the model but it would be interesting to discuss this point further and potentially identify a "better" set of parameter from these portions of flowline.

   **Author response:** Thanks a lot for pointing out the plotting error in panel f) of Fig. 3. The differences in the strain rates are related to a slight shift in the coordinate sampling between the two lines. We apologize for this and will correct it in the revised version of the manuscript.
   As you correctly stated, the differences between the modeled isochrones of flow line A and B in Fig. 4 arise from the different choice of model parameters obtained during the Monte Carlo inversion. As flow line A and B differ in the upstream area, the parameters found are different. This affects the modeled isochrone characteristics but also the parameter sampling in the area where the two lines overlap. It is thus difficult to compare the isochrone fit of the two lines only in the overlapping part since it is also determined by the parameters further upstream. We agree that this should be discussed in the paper and we will insert a few additional sentences in section 4.1.

3. As a general point I think that the limitations of the model should be developed a bit more. The authors state that the lack of radar data is the major limitation of this study but perhaps applying a more evolve 3D model would allow to overcome this issue. I understand that there is a trade off here between using a simple model that allows the Monte Carlo approach and using

a highly under constrained more physically accurate models but the limitation inherent to this choice should be more clearly pointed out.

**Author response:** The main purpose of this study is to determine the source characteristics of ice found in the EastGRIP ice core. As the ice flow along flow-lines is two-dimensional a 3D model is not necessary to determine the source location. Among other 2D models we chose the Dansgaard-Johnsen model due to its few model parameters which make it well suited for the Monte Carlo inversion, as is described in line 151–158. A Monte Carlo inversion of a similar 3D model would be computationally unfeasible as the number of parameters would increase immensely. As you also mention, we state that the largest limitations is the lack of RES data along the EastGRIP flow line. We believe that a 3D model would not overcome this issue as no further information can be obtained without additional constraining data. We will point out the strengths and limitations on our choice of model more clearly in the revised version of the manuscript.

**2 Specific comments**

- Line 5: It is stated here that the location inside an ice stream introduces non climatic bias. But isn't that redundant with the high ice-flow velocities, or is it meant that specific processes linked to the ice stream such as shear margin are causing some of those biases?

  **Author response:** The high ice-flow velocities are mainly responsible for the strong upstream effects at EastGRIP. However, other characteristics of the ice stream could also play a role, for example the higher spatial variability in accumulation rates due to the surface topography (i.e. across the shear margins). In the revised version we will discard the 'due to high ice-flow velocities' as it is included in the 'location in an ice stream'.

- Line 8: It might be worth noting here that the selected flowline emanate from the Greenland Ice Sheet summit but that the site has also specific interest due to the presence of the GRIP ice core.

  **Author response:** We agree that this is a valuable clarification. However, we will mention this in the introduction in order to keep the abstract to the current length.

- Line 9: the RES abbreviation is not re-used in the abstract and could be dropped.

  **Author response:** We agree and will remove the abbreviation.

- Line 10: As the model is solved along a flow line, wouldn't the source be a point rather than an area.

  **Author response:** Yes, we agree that this is confusing. The thought behind the 'area' is that due to uncertainties and the spread between the flow lines we can not determine an exact 'source point'. We will use source location rather than source area in the revised version of the manuscript.

- Line 26: I think that the statement relative to the modelling of NEGIS is not completely true and that more recent modelling work like the ones of Beyer et al. (2018); Smith-Johnsen et al. (2020) should be discussed here.

**Author response:** Thank you for pointing us towards these studies. We have rephrased this section as follows:

**original:**

Understanding the driving mechanisms of the NEGIS is essential to anticipate its future development and potential impact on the ice-sheet stability with large-scale ice-flow models (Joughin et al., 2001; Khan et al., 2014; Vallelonga et al., 2014). Yet, many unknowns remain in our comprehension of ice-stream dynamics (Tulaczyk et al., 2000; Robel et al., 2013), and the underlying processes governing ice flow are not sufficiently understood to successfully reproduce the NEGIS in sophisticated ice-sheet models (e.g. Mottram et al., 2019; Shepherd et al., 2020).

**revised:**

Large-scale ice-sheet models are essential tools to anticipate the future development of the NEGIS and its potential impact on the stability of the GrIS (Joughin et al., 2001; Khan et al., 2014; Vallelonga et al., 2014). However, results obtained from such models often show a significant deviation from observed surface velocitites in the NEGIS and its catchment area (Aschwanden et al., 2016; Mottram et al., 2019). In particular, the high ice flow velocities in the upstream area of the NEGIS and the clearly defined shear margins are difficult to be reproduced by ice flow models (Beyer et al., 2018). A recent study by Smith-Johnsen et al. (2020) showed, that the high surface velocities in the onset region of the ice stream could be reproduced with their model, which however required an exceptionally high and geologically unfeasible geothermal heat flux (Bons et al., 2021). This indicates that additional, yet unknown processes must facilitate ice flow in the NEGIS and that the driving mechanisms governing ice flow are yet not well enough understood.

- Line 27: Some results from the EastGRIP ice core have started to appear but it might be more fair to use the future tense here as much more results are to be produced.

  **Author response:** We will replace 'reveals' with 'by revealing' in order to stress the ongoing process. Later in the introduction we also state that more data will become available in the future.

  **original:**

  The EastGRIP ice core sheds some light on the key processes, as it reveals unique information about ice dynamics, stress regimes, temperatures and basal properties, all of which are crucial components in ice-flow models.

  **revised:**

  The EastGRIP ice core sheds some light on the key processes by revealing unique information about ice dynamics, stress regimes, temperatures and basal properties, all of which are crucial components in ice-flow models.

- Line 37: I am not sure what is referred to here when using the term "lateral flow", is that to contrast with vertical flow? If that is the case perhaps "horizontal flow" would fit better?

  **Author response:** With lateral flow we are referring to the flow away from the central ice

divide. We agree that is unclear and will use 'horizontal flow' in the revised version of the manuscript.

- Line 39: I would prefer "spatial variability of the precipitation".

  **Author response:** We agree and will change it as suggested.

- Line 52: The fact that the model is 2D vertical should be mentioned here.

  **Author response:** Thank you for pointing this out. We will adjust this as follows:

  **original:**

  In this study, we use a two-dimensional Dansgaard–Johnsen model to simulate the ice flow along three approximated flow lines between the ice-sheet summit (GRIP) and EastGRIP.

  **revised:**

  In this study, we use a vertically two-dimensional Dansgaard–Johnsen model to simulate the ice flow along three approximated flow lines between the ice-sheet summit (GRIP) and EastGRIP.

- Line 63: The sentence starting on this line is unclear, perhaps "as a consequence of " should be dropped.

  **Author response:** We agree and will rephrase this part as follows.

  **original:**

  However, as a consequence of error propagation, minor uncertainties and bias in the data severely affect the tracking of flow lines along the velocity field (Hvidberg et al., 2020).

  **revised:**

  Minor uncertainties and bias in these data products strongly affect along-flow tracing and lead to deviations between flow lines derived from different products. These deviations become more pronounced with increasing distance from the starting point, as the uncertainties propagate along the line and in general become larger in slow-moving areas of the ice sheet (Hvidberg et al., 2020).

- Line 67: The deviation of the different flowline does not seem that extreme to me, particularly if they are compared to the spread of the radar lines that are used.

  **Author response:** This is correct. We want to point out here that the flow line we used as the 'present-day' flow line (the black line in Fig.1) includes uncertainties and that it can look different if it is derived from other datasets. We also state that the uncertainties due to missing data is larger than the uncertainties of the flow line itself.

- Figure S1: On this figure I am missing a scale more convenient than the one on the map border which would help to judge distances better on the map.

  **Author response:** Thank you for this great input. We have revised this figure and added a scale. In addition to that, we slightly modified the color map in order to avoid confusion between 'white' velocities and the white background.

- Figure 1: As for Figure S1, Figure 1 would benefit from a more convenient scale legend.

  **Author response:** Same adjustments have been made as for Figure S1.

- Line 79: "the NEGIS trunk"

  **Author response:** We will adapt this suggestion.

- Line 80: Line C also presents a quite substantial data gap and that should be noted here.

  **Author response:** Yes, thank you for pointing this out. We will mention it in the revised version of the manuscript.

- Line 84: "centre" should be capitalised.

  **Author response:** We agree and will adjust it accordingly.

- Line 109: "Greenland Stadial 2 (GS-2)"

  **Author response:** We agree and will introduce the abbreviation for Greenland Stadial here.

- Figure 2: It would be nice to show the different Greenland Stadials on the age axis.

  **Author response:** Thank you a lot for this very valuable input. We decided to indicate the Greenland Stadials in the figure background as colored areas, since the age axis might become too cramped otherwise.

- Line 138: This description of the dating process is not the clearest. I am not sure of what the 250m represent, is that following the radar line up and downstream to smooth out any local bump in the IRH? This whole sentence should probably be rephrased.

  **Author response:** Yes indeed, the depth is averaged over $\pm$ 250 m to smooth out local undulations of the isochrones. We will rephrase this as follows:

  **original:**

  The traced IRHs were dated by assigning the average reflector depth over $\pm$ 250 m around the trace closest to the GRIP and EastGRIP sites to the extended GICC05 time scale (Rasmussen et al., 2014; Seierstad et al., 2014; Mojtabavi et al., 2020).

  **revised:**

  The traced IRHs were dated at both drill sites by assigning the reflector depth at GRIP and EastGRIP to the corresponding time scale. In doing so, local irregularities were smoothed out by averaging the depth over $\pm$ 250 m around the trace closest to the ice core locations.

- Figure 3: On panel b the surface velocity legend is missing

  **Author response:** Yes this is right, thank you a lot for pointing this out. We will adjust it in the revised version of the manuscript.

- Figure 4: I think that "very well" to describe the fit of the isochrones is an over statement. To my eye it seems that there is a bias with the modelled isochrones being slightly higher up in the ice column than the observed one.

**Author response:** We find that the modeled isochrones fit the observed isochrones well within the limitations of the method. Panel b, d and f in Fig. 4. indicate the misfit between the modeled and observed isochrones, which are both, positive and negative. It is true that the simulated isochrones in flow line B tend to be higher than the observed isochrones in the downstream part of the flow line but we do not quite agree that this is the case in general. We will however soften our statement by using 'well' instead of 'very well'.

- Figure 5: I suppose that the basal sliding is expressed as a fraction of the surface velocity. That should be stated in the Figure or in its caption. On panel (i) and (j) the exponent on the x axis is confusingly placed.

  **Author response:** Thank you for the feedback on this figure. We revised it and added the clarification regarding the basal sliding. We also changed the x-axis intervals such that they are identical for each parameter at the same location.

- Figure 6: There is a legend missing in panel (d).

  **Author response:** Thanks for pointing this out. The gray line represents the annual layer thickness obtained from the ice core stratigraphy. The black line is resampled to the same resolution as the modeled layer thicknesses to facilitate comparison. We will revise this figure and add the missing legend.

- Line 295: It is not sure to me what "local accumulation" means in this context. From the rest of the sentence I expect that it is the accumulation at the deposition site but somehow "local" here make it unclear.

  **Author response:** Yes, we are indeed referring to the accumulation rate at the deposition site. We agree that 'local' is confusing in this context and will discard it in the revised version of the manuscript.

- Line 300: The sentence starting on this line is not completely clear. If I refer to the present-day accumulation stated above ($0.12$ ma$^{-1}$) the ($0.14$ma$^{-1}$) given here for interstadial is actually higher then present day.

  **Author response:** Thank you for pointing out the lack of clarity here. The accumulation rates in the glacial period were generally lower than today due to the colder and dryer atmospheric conditions. The ice from that period, however, was deposited further upstream, where the accumulation rate is higher compared to EastGRIP, and hence the accumulation rate at the deposition site was higher than at EastGRIP today. The 'lower' should not refer to the present-day accumulation rate at EastGRIP but to the present-day accumulation rate at the deposition site, see Table 4. We have rephrased this as follows:

  **original:**

  Older ice was, due to climatic reasons, deposited under lower accumulation rates between $0.05$ m a$^{-1}$ in the stadials and $0.14$ m a$^{-1}$ in interstadials.

  **revised:**

  The accumulation rate at the deposition site for older ice varies between $0.05$ m a$^{-1}$ (GS) and

0.14 m a$^{-1}$ (GI). The atmosphere in the glacial period was in general colder and dryer, and hence, accumulation rates were generally lower than today (Cuffey and Clow, 1997). However, due to the upstream flow effects, the ice from the interstadials was deposited under higher accumulation rates than observed at the EastGRIP site today.

- Line 377: The sentence starting on this line should be modified. Zeising and Humbert (2021) actually state in the last part of their paper that "We are aware that these melt rates require an extremely large amount of heat that we suggest to arise from the subglacial water system and the geothermal heat flux." and that they are able to close their energy budget with a more reasonable geothermal heat flux around 0.25 Wm$^{-2}$ .

  **Author response:** We assume that this comment refers to line 357, rather than line 377. We will adjust the sentence as follows:

  **original:**

  Melt rates in these order of magnitudes would require an unusual high geothermal heat flux, immensely exceeding the continental background (Fahnestock et al., 2001; Bons et al., 2021).

  **revised:**

  Melt rates in these order of magnitudes would either require an unusual high geothermal heat flux exceeding the continental background (Fahnestock et al., 2001; Bons et al., 2021) or an additional heat source (Zeising and Humbert, 2021).

- Line 388: "Propagate" might not be the good term here.

  **Author response:** We will replace 'propagate' with 'penetrate'.

- Line 408: It would be nice here to have more information on the reason why this simulation is not attempted. Is it just due to the fact that the constraints on the model would be lacking, that it is not warranted for the specific goal of estimating the source location of the ice or for other reasons.

  **Author response:** The reasoning behind our model choice is described in section 2.3, line 151–158, where the model is introduced. As we mentioned in the answer to the general comment #3, a 3D model is not necessary for the purpose of this study, which is to determine the source location and provide estimates of the past accumulation rates at the deposition site of the EastGRIP ice core. The simple model we are using here has the advantage of having few model parameters which allows the use of the Monte Carlo method. A similar model in 3D would immensely increase computational costs and parameter tuning would become unfeasible. Furthermore, due to the limited availability of radar data, there is not much additional constrains of such a 3D model and hence, the information gain regarding the source location would be very limited. We will revise this section and point out the reasons for not using a 3D model in this context.

**References**

[revised manuscript text omitted]

---

## Author Comment (AC2)

**Authors response on Referee Comment tc-2021-63-RC2**

May 21, 2021

**1 General comments**

Specifically, I find that the methods section is difficult for non-experts to follow, and in particular I find that the current version of the manuscript lacks a description of the authors' logic behind the overall analysis and modeling strategy. This could probably be solved by some rework of each of the method sections to be organized in a logical way as to clearly lead the reader through the authors' reasoning. For example, the addition of a summary paragraph with an introduction of the method steps at the beginning of section 2 and a few sentences at the beginning of each section indicating the purpose of the described methods step would really help ground the reader for each part of the study. Overall, I find that the data and methods section is the most difficult to comprehend, since it seems to lack logical flow. Either adding some more context to lead the reader, or adding something like a flow chart for reference to show the methods graphically, would improve the manuscript greatly. Many of my questions and suggestions stem from vagueness within and my confusion about the written description of the methods.

**Author response:** Thank you for this valuable feedback on the lacking guidance and logical flow throughout the methods section. We will follow your suggestions in the revised version of our manuscript by incorporating a flow-chart to provide an overview of the work-flow leading to our results. We will further introduce a summarizing paragraph in the beginning of section 2 where we intend to outline why the individual steps are necessary and provide the context for the following sections. As you also suggested, we will make the transitions between the subsections smoother and more logic by adding a few sentences providing the context. We hope that this will improve the reading flow and guidance through the text and ultimately improve the quality of the paper.

**2 Specific comments**

- Line 30: Please specify "snow" or "source" deposition

  **Author response:** We will change it to "snow deposition".

- Line 32: "where" → "so" or "therefore" or something similar

  **Author response:** We will replace "where" with "so".

- Line 38: Can you define here what you mean by "non-climatic" within the text? As is, the sentence is confusing since you describe variability in climatic variables as "non-climatic". I think you are technically referring to non "local" climate effects.

**Author response:** Thank you for pointing out the lack of clarity here. We will rephrase this section as follows:

**original:**

The spatial variation in accumulation rate, surface temperature and atmospheric pressure in the upstream area can introduce non-climatic imprints in the ice core (e.g. Koutnik et al., 2016; Fudge et al., 2020).

**revised:**

The spatial variation in accumulation rate, surface temperature and atmospheric pressure in the upstream area can introduce climatic imprints in the ice core record which stem from the advection of ice deposited under different atmospheric conditions. The ice core signal is thus a combination of temporal and spatial variations in climatic components (Koutnik et al., 2016; Fudge et al., 2020) and information on the source characteristics is necessary to interpret ice core measurements within the climatic context.

- Line 40: "parameter" sounds like something from a model instead of the physical characteristic of the ice. Maybe "measurement" or "properties" or something similar?

  **Author response:** Yes, we agree. We will replace 'parameter' with 'quantity' here and make sure that 'parameter' is only used if related to the model throughout the manuscript.

- Line 57: "upstream effects... quantities". It is not clear to me what this sentence means. It would be more impactful if you clearly spelled out and expanded upon for the reader how your results can inform future studies.

  **Author response:** Thank you for pointing out that this is not clear enough. We will rephrase this part as follows:

  **original:**

  The results presented here serve as a basis for corrections of upstream effects in various chemical and physical quantities.

  **revised:**

  The source characteristics presented here form a basis to correct for upstream effects in various chemical and physical quantities of the EastGRIP ice core. These corrections are important to remove any climatic bias in ice-core measurements which are currently analyzed and will become available in the future.

- Line 60: As mentioned above, a brief upper-level description of your methods/strategy would help bring all the below sections together for the reader. In addition, every section in part 2 might also benefit from an introductory sentence or two to describe the motivation for the methods described in the given section and to give context to how it follows from the previously described method steps. Or, even something like a methods flow chart with inputs and outputs as well as the order of your method steps could help convey the information I feel is missing in this section.

**Author response:** Thanks a lot for this input. We will follow your suggestions and implement a flow chart which shows the work flow and guides the reader through this section. We will also add a summarizing paragraph in the beginning of Section 2. The rest of the methods section will be restructured as follows: 2.1) Description of the flow line derivation and the selection of RES images approximating the flow line derived from satellite based surface velocities. 2.2) Extension of the EastGRIP chronology. 2.3) Tracing and dating of the isochrones. 2.4) Full description of the ice flow model, including the climate model. 2.5) Expanded description on the Monte Carlo method.

- Line 72: I suggest that the information in paragraph 3 of this section (line 3) introducing the radar data be placed before paragraph 2 here, to add context to the flightline discussion.

  **Author response:** Thank you for this suggestion. We will move the information from line 119-126 to section 2.1.

- Lines 83 and 86: References to paper in preparation are not appropriate.

  **Author response:** The referenced paper is available as preprint now and will be cited accordingly.

- Lines 94-96: I think you are saying here that the Mojtabavi et al. (2020) paper is motivating and informing this work, and for this study you extend the analysis through the depth reached by 2019 (which is from 49.ka b2k). But it took me many times reading these sentences to understand that relationship. Please try to rephrase so that the relationship between the past work and what is done is this study is clear to the reader (especially readers who might not be experts in ice core analysis).

  **Author response:** Thank you for pointing out that this relationship is unclear. We have built up on the work by Mojtabavi et al. (2020) and extended their timescale to the current EastGRIP depth in order to date the isochrones observed in the radargrams. We will rephrase this section as follows:

  **original:**

  Mojtabavi et al. (2020) synchronized the EastGRIP and NorthGRIP ice cores for the last 15 kyr in order to apply the Greenland Ice Core Chronology 2005 (GICC05 Andersen et al., 2006) to EastGRIP. By 2019, the ice-core drilling progressed down to 2,122.45 m, allowing us to extend the time scale to 49.2 ka b2k (thousands of years before 2000 CE).

  **revised:**

  The validation of our modeling results and the correct dating of isochrones requires a reliable depth-age scale. The Greenland Ice Core Chronology 2005 (GICC05 Vinther et al., 2006; Rasmussen et al., 2006; Andersen et al., 2006; Svensson et al., 2006) is based on annual layer counting in various Greenland ice cores. It has been transferred to GRIP and other deep drilling sites in Greenland by synchronizing the ice cores with each other using horizons of e.g. volcanic origin (Rasmussen et al., 2008; Seierstad et al., 2014). The upper 1,383.84 m of the EastGRIP ice core were drilled between 2015 and 2018, and synchronized with the NorthGRIP ice core in previous work by Mojtabavi et al. (2020) to transfer the GICC05 chronology to EastGRIP.

By 2019, the ice-core drilling progressed down to 2,122.45 m, allowing us to extend the existing time scale from 15 ka to 49.9 ka b2k (thousands of years before 2000 CE). As part of the present study we identified common isochrones between EastGRIP, NorthGRIP and NEEM to transfer the GICC05 chronology to the part of the EastGRIP record which is not yet synchronized. This involved the same methods applied to NEEM by Rasmussen et al. (2013) and to the upper 1,383.84 m of EastGRIP by Mojtabavi et al. (2020).

- Line 104: "termination of the Greenland Stadial (GS) 2"

  **Author response:** We will introduce the missing abbreviation here, thank you for noticing it.

- Line 118: As for the other sections, some summary sentences motivating why you need to do this step and the context of how it informs the following steps would be highly appreciated. The way it is written now, that knowledge is assumed and the transition from the previous section to this section is very abrupt.

  **Author response:** Thanks again for this input. We agree that this section requires some restructuring and smoother transitions to ensure the reading flow and guidance of readers outside of the ice-core community. We will add the following information in the beginning of the section:

  The depth-age relationship from ice core chronologies can be extended in the lateral plane by tracing and dating of isochrones in RES images. The depth of these isochrones along the EastGRIP flow lines is part of the observed data used to tune the ice flow model parameters in the Monte Carlo inversion.

- Line 127: Which Matlab program is used?

  **Author response:** The Matlab program we used is called "picking tool" and was developed by Aslak Grinsted. It is not publicly available and we will mention it in the revised version of the manuscript. Thank you for pointing it out.

- Lines 129: "subsequent" → "subsequently"

  **Author response:** We agree.

- Lines 128-130: I am not sure what this sentence means. Please try to rephrase it so that it more clearly describes the method used.

  **Author response:** We agree that the description is not the clearest and will rephrase it as follows:

  **original**:

  The algorithm is based on calculating the local slope in each pixel of the RES image by minimizing the variance along a local line segment. Layers are traced automatically between two user-defined points by following the steepest slope from both ends and subsequent weighting of the two lines by distance to the end points. The number of picks required for thorough tracing depends on the data quality and reflector strength.

  **revised:**

The program is based on calculating the local slope in each pixel of the RES image, and layers are traced automatically between two user-defined points. Starting from each of these points the algorithm walks along the steepest slope towards the other point. Subsequently, the two lines are weighted by distance to their starting point and combined to one layer. The number of picks required for thorough tracing depends on the data quality and reflector strength.

- Line 156: "propagation" and also their evolution (i.e. thickness)?

  **Author response:** We prefer to use 'deformation' instead of 'evolution' and will rephrase the sentence as follows:

  **original:**

  Here, we use a two-dimensional Dansgaard–Johnsen model (Dansgaard and Johnsen, 1969) to simulate the propagation of internal layers along approximated flow lines between the ice-sheet summit (GRIP) and EastGRIP.

  **revised:**

  Here, we use a two-dimensional Dansgaard–Johnsen model (Dansgaard and Johnsen, 1969) to simulate the propagation **and deformation** of internal layers along approximated flow lines between the ice-sheet summit (GRIP) and EastGRIP.

- Line 164: Use of "B" for base is confusing because it was previously used in eq. 2 as radar bandwidth.

  **Author response:** Many thanks for pointing this out. We will revise the notation throughout the manuscript and use subscript 'bed' instead of 'B'.

- Line 172: Please define w within the text, e.g. "vertical velocities (w)"

  **Author response:** We agree and will make the suggested adjustment.

- Line 175: Please define S and B in the text (e.g. "surface (S) and bedrock (B) are:"

  **Author response:** We agree and will define the parameters in the text. To avoid confusion with the misfit (S) and radar bandwidth (B) we will use $E_{sur}$ and $E_{bed}$ for surface and bed elevation instead.

- Line 183: I suggest adding → age of the isochrones "above the bed, respectively".

  **Author response:** We agree and will add 'above the bed' for clarification.

- Line 199: Please define what is mean by the "slope" of the accumulation

  **Author response:** The 'slope' comes from the parameter definition, which is the relative slopes of the accumulation rate in warm and cold climate, i.e. they describe how sensitive the accumulation is towards changes in the $\delta^{18}O$ record. We agree that the word 'slope' is confusing and will rephrase the sentence. We also realized that a reference towards the previous studies of Grinsted and Dahl-Jensen (2002) and Buchardt and Dahl-Jensen (2007) which used a similar model is appropriate here. We will revise this part as follows:

  **original:**

The unknown parameters $c_w$ and $c_c$ are defined as the relative slopes of the accumulation rates in warm ($c_w$) and cold ($c_c$) periods:

**revised:**

The parameters $c_w$ and $c_c$ determine the sensitivity of the accumulation rate with varying $\delta^{18}O$ in warm ($c_w$) and cold ($c_c$) periods and are defined as (Grinsted and Dahl-Jensen, 2002; Buchardt and Dahl-Jensen, 2007):

- Line 202: Please quantify or more clearly describe what is considered a "good" approximation

  **Author response:** We evaluated the parameters $c_w$ and $c_c$ by trial and error of the initial model and found that the isochrone misfit was smallest for $c_w = 0.1$ and $c_c = 0.15$. To better constrain these parameters in the revised version of the manuscript, we used $c_w = 0.1$ and $c_c = 0.15$ as initial parameters and further tuned them with the Monte Carlo method, together with the other model parameters.

- Line 203: This methods section is the most difficult to understand. It would improve the manuscript if it were expanded and reworked to be clearer (and reproducible by future studies).

  **Author response:** We agree that this section should be elaborated more and we will expand it in the revised version of the manuscript. Specifically, we will add more information on the Monte Carlo method and why it is used in this context, describe more precisely the initial model assumptions and parameter regularization, and elaborate in detail how the model parameters are perturbed. The individual steps of the Monte Carlo iterations will be displayed in the flow-chart for additional guidance.

- Line 204: How are your parameters initialized?

  **Author response:** We described the initialization of the model parameters in section 2.3, line 184–190, but we agree that it would make more sense here. We will thus move the description of the parameter initialization from Sect. 2.3 to Sect. 2.5.

- Line 205: Please specify here how the eight are selected (e.g. reference to Table 3, equally spaced for computational efficiency, etc)

  **Author response:** We agree that a reference to Table 3 would be appropriate here and will incorporate this in the revised version. We will also move the reasoning of why only eight isochrones are used from line 240 to this section. The reason for choosing the specific layers is a mixture between achieving approximately equal vertical spacing and good agreement between the ages obtained at the GRIP and EastGRIP sites. We will also mention this in the revised version of the manuscript.

- Eq. 18: The use of S is confusing because it was used to mean surface in earlier equations. Also, please define in the text (i.e. "The misfit (S)").

  **Author response:** Thank you for pointing this out. As mentioned in a previous comment we will look over the notations to make sure that the symbols are unique and we will define the quantities in the text.

- Line 208: Please define in this context what is meant by uncertainty. Is it the standard deviation or a defined uniform error spread? How is it determined?

  **Author response:** Thank you for this feedback. The data uncertainty is assumed to be the standard deviation of a gaussian distribution. The uncertainty of the isochrone depth is 13 m, equivalent to the maximum depth uncertainty related to the picking process and radar range resolution. We will add this information and describe the definition of the misfit more thoroughly in the revised version of the manuscript.

- Line 209: Please quantify "a large number"

  **Author response:** This is very difficult to quantify because the probability density in the model space is not known. In theory, there is an infinite amount of solutions and, hence, the probability density can contain an infinite amount of local maxima. The advantage of the Monte Carlo method over local search methods is that it is less likely to get trapped in one of the local maxima. We have rewritten this entire section to be more clear on the reasoning behind the used method.

- Line 214: "mcurr is perturbed" $\rightarrow$ Please specify how the parameters themselves are perturbed. For example, are they all perturbed independently from each other?

  **Author response:** Thank you for pointing out this missing explanation. We perturb one (randomly selected) parameter per iteration, independent from each other. We will elaborate this further in the revised version of the manuscript by adding a paragraph describing in detail how this is done.

- Line 218: What is meant by a "burn-in phase"? Please specify in the text.

  **Author response:** The term 'burn-in phase' is commonly used in Markov Chain Monte Carlo methods and refers to the initial phase of the sampling process, where the model moves from the initial state to a high-probability area. Models sampled during this initial phase should be discarded to ensure the sampling of the target probability. We will formulate this more explicitly in the revised version of the manuscript.

- Line 219: Please specify in the text how the thresholds (or maximum deviations) are determined?

  **Author response:** Thanks for pointing out the lack of information here. We defined the threshold as follows:

  - The kink height is allowed to vary within the ice thickness, i.e. $0 < h < H$. Anything outside this range would physically not make sense. We did not regulate this parameter further because we do not have any prior knowledge on the vertical velocity distribution along the flow line.

  - The accumulation rate was set to vary within $\pm 2$ cm from the initial accumulation rate, i.e. $\lambda_{H,0} - 0.02 < \lambda_H < \lambda_{H,0} + 0.02$. The initial accumulation rate obtained from Eq. (12) fits quite well with field observations at GRIP and EastGRIP. Nevertheless, small deviations can be expected because the local layer approximation is not valid in the study area. We estimate that this deviation should be within $\pm 2$ cm from the initial guess.

- The basal sliding was allowed to vary $\pm$ 15 % from the initial parameterization, i.e. $f_{B,0} - 0.15 < f_B < f_{B,0} + 0.15$, resulting in a maximum sliding fraction of 95 % in the vicinity of EastGRIP.

- The threshold for the basal melt rate was set to 0.03 m a$^{-1}$, i.e. $\lambda_{B,0} - 0.03 < \lambda_B < \lambda_{B,0} + 0.03$ which allows a maximum melt rate of 0.06 m a$^{-1}$ at EastGRIP. The choice of this upper limit is based on preliminary modeling results which showed that much higher basal melt rates lead to erosion of the deepest observed isochrones.

As pointed out by Referee # 1, the basal melt rates are reaching the limits at EastGRIP for flow line A and B, and the upper threshold is lower than values suggested in this area in previous studies (e.g. Fahnestock et al., 2001; Keisling et al., 2014; Zeising and Humbert, 2021). We will thus repeat our model runs allowing the parameters to be more free in the revised version of the Manuscript.

- Figure 3 (caption), Line 5: "(a,e,j)" → "(a,e,i)" ?

**Author response:** Yes, this is correct. Many thanks for pointing this out.

- Figure 3 (caption): The particle trajectories should probably be described earlier in the caption when panels a,e, and i are first described. At this point, please distinguish that they are illustrated by the solid lines and that the IRHs are represented by broken lines of the same color to indicate age of deposition.

**Author response:** Thank you for this valuable suggestion. We will move the description of particle trajectories further up in the caption. We will also add clarification on what the different line styles and the colors indicate.

- Figure 5: Please add a label for the y axis (i.e. number of model scenarios/samples/runs or something more appropriate).

**Author response:** Thank you for pointing this out. We have revised this figure and added '# of samples' on the y-axes.

- Table 4: Could uncertainties also be included in this table?

**Author response:** This is a good idea, thank you for this suggestion. We will include the maximum spread between the different flow lines in the table.

- Figure 6: It appears that the colored lines for panel f are colored with the same scheme as the lines in Fig. 3 and Fig. 4. This is a very nice feature and connection between the figures, and I don't remember this being strongly noted in the text. I may have missed it, but if was not, please make sure to point out to the reader that you made this effort, since it is a definitely a helpful tool. For example, in Fig. 3 and Fig. 4 captions you could reference Fig. 6f as to where one could see where the lines fall within the core. It would also be helpful if this nice connection was pointed out clearly in the text. In addition, for Fig. 6, maybe you could highlight in the panel f somehow the 8 chosen isochrones or connect them with their layer number as noted in Table 3.

**Author response:** Thank you for this great input. We did not mention it in the text but will

do so in the revised version of the manuscript. We will also follow your suggestion of referring to this figure in Fig. 3 and Fig. 4.

- Line 298: Please reference a figure (i.e. Fig. 3c,g,k). Also, add directionality to this statement for clarity. i.e. "increasing accumulation along the flow line with distance upstream" or something similar

  **Author response:** Thank you for pointing this out. We will refer to Fig 3 and 4 in the revised version of the manuscript. We will also add the directionality and will make sure that it is clear in all similar statements throughout the manuscript.

- Line 300: It would be helpful to directly specify what exactly the climatic reasons are for this.

  **Author response:** Yes, indeed. We will mention here that this is related to the colder and dryer atmospheric conditions (Cuffey and Clow, 1997).

- Lines 301-302: This sentence is awkward and could be rephrased to be clearer as to lead the reader more directly. Please also reference the figures and panels that support your argument.

  **Author response:** We agree that this part should be rephrased. We will also refer to the corresponding figure panels in the revised version of the manuscript. The planned adjustments look as follows:

  **original:**

  Older ice was, due to climatic reasons, deposited under lower accumulation rates between 0.05 m a$^{-1}$ in the stadials and 0.14 m a$^{-1}$ in interstadials.
  The accumulation-rate variations between the three flow lines are a combination of the varying along-flow accumulation pattern and upstream distance of the source area, and the model spread provides important uncertainty estimates.

  **revised:**

  The accumulation rate at the deposition site for older ice varies between 0.05 m a$^{-1}$ (GS) and 0.14 m a$^{-1}$ (GI). The atmosphere in the glacial period was in general colder and dryer, and hence, accumulation rates were generally lower than today (Cuffey and Clow, 1997). However, due to the upstream flow effects, the ice from the interstadials was deposited under higher accumulation rates than observed at the EastGRIP site today. The variations in the past accumulation-rates between the three flow lines result from both, the varying along-flow accumulation pattern and different upstream distance of the source location. The spread between the three models provides important uncertainty estimates.

- Line 333: Please be clearer about what is meant by "out-of-plane effects" and why this means that the core is not affected.

  **Author response:** It means that the way these isochrones are deformed is not directly related to the ice flow along the flow-line since this area deviates considerably from the surface flow direction. The undulations observed here could for instance be related to features of the bed topography east of this flight line as that's where the ice most likely originates. The ice we find in the EastGRIP ice core is unlikely to have passed this area and therefore it is not so important

that these undulations can not be reproduced by our model. We will try to specify this more clearly in the revised version of the manuscript.

**original:**

We argue that these strongly deformed isochrones are out-of-the-plane effects since they predominantly appear in parts of the flow lines which deviate from the observed surface velocity direction by more than 15 degrees. Accordingly, the ice in the EastGRIP ice core is presumably not affected by them, and the fact that they are not reproduced by the model does not put any constraints on the usefulness of our results for upstream corrections.

**revised:**

These strongly deformed isochrones predominantly appear in parts of the flow lines which deviate from the observed surface velocity direction by more than 15 degrees. We thus argue that they are out-off-the-plane effects and that the isochrones parallel to the ice flow direction are not as strongly deformed. Accordingly, the ice in the EastGRIP ice core has not experienced such deformation, and the fact that they are not reproduced by the model does not put any constraints on the usefulness of our results for upstream corrections.

- Line 429: our model allowed "us" to invert for

  **Author response:** We agree and will adjust this in the revised version of the manuscript.

- Line 436: As for line 298 comment above please be specific about directionality of the increase in accumulation.

  **Author response:** Again, a very important point, thanks for making us aware of it. As mentioned above we will address this and check the entire manuscript for similar unclear statements in terms of directionality.

**References**

[revised manuscript text omitted]